# Partial Hard Thresholding: Towards A Principled Analysis of Support Recovery

**Jie Shen**
Department of Computer Science
School of Arts and Sciences
Rutgers University
New Jersey, USA
js2007@rutgers.edu

**Ping Li**
Department of Statistics and Biostatistics
Department of Computer Science
Rutgers University
New Jersey, USA
pingli@stat.rutgers.edu

## Abstract

In machine learning and compressed sensing, it is of central importance to understand when a tractable algorithm recovers the support of a sparse signal from its compressed measurements. In this paper, we present a principled analysis on the support recovery performance for a family of hard thresholding algorithms. To this end, we appeal to the partial hard thresholding (PHT) operator proposed recently by Jain et al. [IEEE Trans. Information Theory, 2017]. We show that under proper conditions, PHT recovers an arbitrary $s$-sparse signal within $O(s\kappa \log \kappa)$ iterations where $\kappa$ is an appropriate condition number. Specifying the PHT operator, we obtain the best known results for hard thresholding pursuit and orthogonal matching pursuit with replacement. Experiments on the simulated data complement our theoretical findings and also illustrate the effectiveness of PHT.

## 1 Introduction

This paper is concerned with the problem of recovering an *arbitrary* sparse signal from a set of its (compressed) measurements. We say that a signal $\bar{x} \in \mathbb{R}^d$ is $s$-sparse if there are no more than $s$ non-zeros in $\bar{x}$. This problem, together with its many variants, have found a variety of successful applications in compressed sensing, machine learning and statistics. Of particular interest is the setting where $\bar{x}$ is the true signal and only a small number of linear measurements are given, referred to as compressed sensing. Such instance has been exhaustively studied in the last decade, along with a large body of elegant work devoted to efficient algorithms including $\ell_1$-based convex optimization and hard thresholding based greedy pursuits [7, 6, 15, 8, 3, 5, 11]. Another quintessential example is the sparsity-constrained minimization program recently considered in machine learning [30, 2, 14, 22], for which the goal is to efficiently learn the global sparse minimizer $\bar{x}$ from a set of training data. Though in most cases, the underlying signal can be categorized into either of the two classes, we note that it could also be other object such as the parameter of logistic regression [19]. Hence, for a unified analysis, this paper copes with an arbitrary sparse signal and the results to be established quickly apply to the special instances above.

It is also worth mentioning that while one can characterize the performance of an algorithm and can evaluate the obtained estimate from various aspects, we are specifically interested in the quality of support recovery. Recall that for sparse recovery problems, there are two prominent metrics: the $\ell_2$ distance and the support recovery. Theoretical results phrased in terms of the $\ell_2$ metric is also referred to as parameter estimation, on which most of the previous papers emphasized. Under this metric, many popular algorithms, e.g., the Lasso [24, 27] and hard thresholding based algorithms [9, 3, 15, 8, 10, 22], are guaranteed with accurate approximation up to the energy of noise. Support recovery is another important factor to evaluate an algorithm, which is also known as feature

selection or variable selection. As one of the earliest work, [25] offered sufficient and necessary conditions under which orthogonal matching pursuit and basis pursuit identify the support. The theory was then developed by [35, 32, 27] for the Lasso estimator and by [29] for the garrotte estimator.

Typically, recovering the support of a target signal is more challenging than parameter estimation. For instance, [18] showed that the restricted eigenvalue condition suffices for the Lasso to produce an accurate estimate whereas in order to recover the sign pattern, a more stringent mutual incoherence condition has to be imposed [27]. However, as has been recognized, if the support is detected precisely by a method, then the solution admits the optimal statistical rate [27]. In this regard, research on support recovery continues to be a central theme in recent years [33, 34, 31, 4, 17]. Our work follows this line and studies the support recovery performance of hard thresholding based algorithms, which enjoy superior computational efficiency to the convex programs when manipulating a huge volume of data [26].

We note that though [31, 4] have carried out theoretical understanding for hard thresholding pursuit (HTP) [10], showing that HTP identifies the support of a signal within a few iterations, neither of them obtained the general results in this paper. To be more detailed, under the restricted isometry property (RIP) condition [6], our iteration bound holds for an arbitrary sparse signal of interest, while the results from [31, 4] hold either for the global sparse minimizer or for the true sparse signal. Using a relaxed sparsity condition, we obtain a clear iteration complexity $O(s\kappa \log \kappa)$ where $\kappa$ is a proper condition number. In contrast, it is hard to quantify the bound of [31] (see Theorem 3 therein). From the algorithmic perspective, we consider a more general algorithm than HTP. In fact, we appeal to the recently proposed partial hard thresholding (PHT) operator [13] and demonstrate novel results, which in turn indicates the best known iteration complexity for HTP and orthogonal matching pursuit with replacement (OMPR) [12]. Thereby, the results in this paper considerably extend our earlier work on HTP [23]. It is also worth mentioning that, though our analysis hinges on the PHT operator, the support recovery results to be established are stronger than the results in [13] since they only showed parameter estimation of PHT. Finally, we remark that while a couple of previous work considered signals that are not exactly sparse (e.g., [4]), we in this paper focus on the sparse case. Extensions to the generic signals are left as interesting future directions.

**Contribution.** The contribution of this paper is summarized as follows. We study the iteration complexity of the PHT algorithm, and show that under the RIP condition or the relaxed sparsity condition (to be clarified), PHT recovers the support of an arbitrary $s$-sparse signal within $O(s\kappa \log \kappa)$ iterations. This strengthens the theoretical results of [13] where only parameter estimation of PHT was established. Thanks to the generality of the PHT operator, our results shed light on the support recovery performance of a family of prevalent iterative algorithms. As two extreme cases of PHT, the new results immediately apply to HTP and OMPR, and imply the best known bound.

**Roadmap.** The remainder of the paper is organized as follows. We describe the problem setting, as well as the partial hard thresholding operator in Section 2, followed by the main results regarding the iteration complexity. In Section 3, we sketch the proof of the main results and list some useful lemmas which might be of independent interest. Numerical results are illustrated in Section 4 and Section 5 concludes the paper and poses several interesting future work. The detailed proof of our theoretical results is deferred to the appendix (see the supplementary file).

**Notation.** We collect the notation that is involved in this paper. The upper-right letter C and its subscript variants (e.g., $C_1$) are used to denote absolute constants whose values may change from appearance to appearance. For a vector $\boldsymbol{x} \in \mathbb{R}^d$, its $\ell_2$ norm is denoted by $\|\boldsymbol{x}\|$. The support set of $\boldsymbol{x}$ is denoted by $\mathrm{supp}\,(\boldsymbol{x})$ which indexes the non-zeros in $\boldsymbol{x}$. With a slight abuse, $\mathrm{supp}\,(\boldsymbol{x}, k)$ is the set of indices for the $k$ largest (in magnitude) elements. Ties are broken lexicographically. We interchangeably write $\|\boldsymbol{x}\|_0$ or $|\mathrm{supp}\,(\boldsymbol{x})|$ to signify the cardinality of $\mathrm{supp}\,(\boldsymbol{x})$. We will also consider a vector restricted on a support set. That is, for a $d$-dimensional vector $\boldsymbol{x}$ and a support set $T \subset \{1, 2, \ldots, d\}$, depending on the context, $\boldsymbol{x}_T$ can either be a $|T|$-dimensional vector by extracting the elements belonging to $T$ or a $d$-dimensional vector by setting the elements outside $T$ to zero. The complement of a set $T$ is denoted by $\overline{T}$.

We reserve $\bar{\boldsymbol{x}} \in \mathbb{R}^d$ for the target $s$-sparse signal whose support is denoted by $S$. The quantity $\bar{\boldsymbol{x}}_{\min} > 0$ is the minimum absolute element in $\bar{\boldsymbol{x}}_S$, where we recall that $\bar{\boldsymbol{x}}_S \in \mathbb{R}^s$ consists of the non-zeros of $\bar{\boldsymbol{x}}$. The PHT algorithm will depend on a carefully chosen function $F(\boldsymbol{x})$. We write its gradient as $\nabla F(\boldsymbol{x})$ and we use $\nabla_k F(\boldsymbol{x})$ as a shorthand of $(\nabla F(\boldsymbol{x}))_{\mathrm{supp}(\nabla F(\boldsymbol{x}),k)}$, i.e., the top $k$ absolute components of $\nabla F(\boldsymbol{x})$.

## 2 Partial Hard Thresholding

To pursue a sparse solution, hard thresholding has been broadly invoked by many popular greedy algorithms. In the present work, we are interested in the partial hard thresholding operator which sheds light upon a unified design and analysis for iterative algorithms employing this operator and the hard thresholding operator [13]. Formally, given a support set $T$ and a freedom parameter $r > 0$, the PHT operator which is used to produce a $k$-sparse approximation to $z$ is defined as follows:

$$\text{PHT}_k\left(\boldsymbol{z}; T, r\right) := \underset{\boldsymbol{x}\in\mathbb{R}^d}{\arg\min} \|\boldsymbol{x} - \boldsymbol{z}\|, \text{ s.t. } \|\boldsymbol{x}\|_0 \leq k, \ |T \setminus \text{supp}\left(\boldsymbol{x}\right)| \leq r. \tag{1}$$

The first constraint simply enforces a $k$-sparse solution. To gain intuition on the second one, consider that $T$ is the support set of the last iterate of an iterative algorithm, for which $|T| \leq k$. Then the second constraint ensures that the new support set differs from the previous one by at most $r$ positions. As a special case, one may have noticed that the PHT operator reduces to the standard hard thresholding when picking the freedom parameter $r \geq k$. On the other spectrum, if we look at the case where $r = 1$, the PHT operator yields the interesting algorithm termed orthogonal matching pursuit with replacement [12], which in general replaces one element in each iteration.

It has been shown in [13] that the PHT operator can be computed in an efficient manner for a general support set $T$ and a freedom parameter $r$. In this paper, our major focus will be on the case $|T| = k$[1]. Then Lemma 1 of [13] indicates that $\text{PHT}_k\left(\boldsymbol{z}; T, r\right)$ is given as follows:

$$\text{top} = \text{supp}\left(\boldsymbol{z}_{\overline{T}}, r\right), \ \text{PHT}_k\left(\boldsymbol{z}; T, r\right) = \text{HT}_k\left(\boldsymbol{z}_{T\cup\text{top}}\right), \tag{2}$$

where $\text{HT}_k(\cdot)$ is the standard hard thresholding operator that sets all but the $k$ largest absolute components of a vector to zero.

Equipped with the PHT operator, we are now in the position to describe a general iterative greedy algorithm, termed $\text{PHT}(r)$ where $r$ is the freedom parameter in (1). At the $t$-th iteration, the algorithm reveals the last iterate $\boldsymbol{x}^{t-1}$ as well as its support set $S^{t-1}$, and returns a new solution as follows:

$$\boldsymbol{z}^t = \boldsymbol{x}^{t-1} - \eta\nabla F(\boldsymbol{x}^{t-1}),$$
$$\boldsymbol{y}^t = \text{PHT}_k\left(\boldsymbol{z}^t; S^{t-1}, r\right), \ S^t = \text{supp}\left(\boldsymbol{y}^t\right),$$
$$\boldsymbol{x}^t = \underset{\boldsymbol{x}\in\mathbb{R}^d}{\arg\min} \ F(\boldsymbol{x}), \text{ s.t. } \text{supp}\left(\boldsymbol{x}\right) \subset S^t.$$

Above, we note that $\eta > 0$ is a step size and $F(\boldsymbol{x})$ is a proxy function which should be carefully chosen (to be clarified later). Typically, the sparsity parameter $k$ equals $s$, the sparsity of the target signal $\bar{\boldsymbol{x}}$. In this paper, we consider a more general choice of $k$ which leads to novel results. For further clarity, several comments on $F(\boldsymbol{x})$ are in order.

First, one may have observed that in the context of sparsity-constrained minimization, the proxy function $F(\boldsymbol{x})$ used above is chosen as the objective function [30, 14]. In that scenario, the target signal is a global optimum and $\text{PHT}(r)$ proceeds as projected gradient descent. Nevertheless, recall that our goal is to estimate an arbitrary signal $\bar{\boldsymbol{x}}$. It is not realistic to look for a function $F(\boldsymbol{x})$ such that our target happens to be its global minimizer. The remedy we will offer is characterizing a deterministic condition between $\bar{\boldsymbol{x}}$ and $\nabla F(\bar{\boldsymbol{x}})$ which is analogous to the signal-to-noise ratio condition, so that any function $F(\boldsymbol{x})$ fulfilling that condition suffices. In this light, we find that $F(\boldsymbol{x})$ behaves more like a proxy that guides the algorithm to a given target. Remarkably, our analysis also encompasses the situation considered in [30, 14].

Second, though it is not being made explicitly, one should think of $F(\boldsymbol{x})$ as containing the measurements or the training data. Consider, for example, recovering $\bar{\boldsymbol{x}}$ from $\boldsymbol{y} = \boldsymbol{A}\bar{\boldsymbol{x}}$ where $\boldsymbol{A}$ is a design matrix and $\boldsymbol{y}$ is the response (both are known). A natural way would be running the $\text{PHT}(r)$ algorithm with $F(\boldsymbol{x}) = \|\boldsymbol{y} - \boldsymbol{A}\boldsymbol{x}\|^2$. One may also think of the logistic regression model where $\boldsymbol{y}$ is a binary vector (label), $\boldsymbol{A}$ is a collection of training data (feature), and $F(\boldsymbol{x})$ is the logistic loss evaluated on the training samples.

With the above clarification, we are ready to make assumptions on $F(\boldsymbol{x})$. It turns out that two properties of $F(\boldsymbol{x})$ are vital for our analysis: restricted strong convexity and restricted smoothness. These two conditions were proposed by [16] and have been standard in the literature [34, 1, 14, 22].

**Definition 1.** We say that a differentiable function $F(\boldsymbol{x})$ satisfies the property of restricted strong convexity (RSC) at sparsity level $s$ with parameter $\rho_s^- > 0$ if for all $\boldsymbol{x}, \boldsymbol{x}' \in \mathbb{R}^d$ with $\|\boldsymbol{x} - \boldsymbol{x}'\|_0 \leq s$,

$$F(\boldsymbol{x}) - F(\boldsymbol{x}') - \langle \nabla F(\boldsymbol{x}'), \boldsymbol{x} - \boldsymbol{x}' \rangle \geq \frac{\rho_s^-}{2} \|\boldsymbol{x} - \boldsymbol{x}'\|^2.$$

Likewise, we say that $F(\boldsymbol{x})$ satisfies the property of restricted smoothness (RSS) at sparsity level $s$ with parameter $\rho_s^+ > 0$ if for all $\boldsymbol{x}, \boldsymbol{x}' \in \mathbb{R}^d$ with $\|\boldsymbol{x} - \boldsymbol{x}'\|_0 \leq s$, it holds that

$$F(\boldsymbol{x}) - F(\boldsymbol{x}') - \langle \nabla F(\boldsymbol{x}'), \boldsymbol{x} - \boldsymbol{x}' \rangle \leq \frac{\rho_s^+}{2} \|\boldsymbol{x} - \boldsymbol{x}'\|^2.$$

We call $\kappa_s = \rho_s^+ / \rho_s^-$ as the condition number of the problem, since it is essentially identical to the condition number of the Hessian matrix of $F(\boldsymbol{x})$ restricted on $s$-sparse directions.

## 2.1 Deterministic Analysis

The following proposition shows that under very mild conditions, PHT($r$) either terminates or recovers the support of an arbitrary $s$-sparse signal $\bar{\boldsymbol{x}}$ using at most $O(s\kappa_{2s} \log \kappa_{2s})$ iterations.

**Proposition 2.** *Consider the PHT($r$) algorithm with $k = s$. Suppose that $F(\boldsymbol{x})$ is $\rho_{2s}^-$-RSC and $\rho_{2s}^+$-RSS, and the step size $\eta \in (0, 1/\rho_{2s}^+)$. Let $\kappa := \rho_{2s}^+ / \rho_{2s}^-$. Then PHT($r$) either terminates or recovers the support of $\bar{\boldsymbol{x}}$ within $O(s\kappa \log \kappa)$ iterations provided that $\bar{\boldsymbol{x}}_{\min} \geq \frac{4\sqrt{2} + 2\sqrt{\kappa}}{\rho_{2s}^-} \|\nabla_{2s} F(\bar{\boldsymbol{x}})\|.$*

A few remarks are in order. First, we remind the reader that under the conditions stated above, it is *not* guaranteed that PHT($r$) succeeds. We say that PHT($r$) fails if it terminates at some time stamp $t$ but $S^t \neq S$. This indeed happens if, for example, we feed it with a bad initial point and pick a very small step size. In particular, if $\boldsymbol{x}_{\min}^0 > \eta \|\nabla F(\boldsymbol{x}^0)\|_\infty$, then the algorithm makes no progress. The crux to remedy this issue is imposing a lower bound on $\eta$ or looking at more coordinates in each iteration, which is the theme below. However, the proposition is still useful because it asserts that as far as we make sure that PHT($r$) runs long enough (i.e., $O(s\kappa \log \kappa)$ iterations), it recovers the support of an arbitrary sparse signal. We also note that neither the RIP condition nor a relaxed sparsity is assumed in this proposition.

The $\bar{\boldsymbol{x}}_{\min}$-condition above is natural, which can be viewed as a generalization of the well-known signal-to-noise ratio (SNR) condition. This follows by considering the noisy compressed sensing problem, where $\boldsymbol{y} = \boldsymbol{A}\bar{\boldsymbol{x}} + \boldsymbol{e}$ and $F(\boldsymbol{x}) = \|\boldsymbol{y} - \boldsymbol{A}\boldsymbol{x}\|^2$. Here, the vector $\boldsymbol{e}$ is some noise. Now the RSC and RSS imply for any $2s$-sparse $\boldsymbol{x}$

$$\rho_{2s}^- \|\boldsymbol{x}\|^2 \leq \|\boldsymbol{A}\boldsymbol{x}\|^2 \leq \rho_{2s}^+ \|\boldsymbol{x}\|^2.$$

Hence

$$\|\nabla_{2s} F(\bar{\boldsymbol{x}})\| = \left\|(\boldsymbol{A}^\top \boldsymbol{e})_{2s}\right\| = \Theta(\|\boldsymbol{e}\|)$$

In fact, the $\bar{\boldsymbol{x}}_{\min}$-condition has been used many times to establish support recovery. See, for example, [31, 4, 23].

In the following, we strengthen Prop. 2 by considering the RIP condition which requires a well-bounded condition number, i.e., $\kappa \leq O(1)$.

**Theorem 3.** *Consider the PHT($r$) algorithm with $k = s$. Suppose that $F(\boldsymbol{x})$ is $\rho_{2s+r}^-$-RSC and $\rho_{2s+r}^+$-RSS. Let $\kappa := \rho_{2s+r}^+ / \rho_{2s+r}^-$ be the condition number which is smaller than $1 + 1/(\sqrt{2} + \nu)$ where $\nu = \sqrt{s - r + 2}$. Pick the step size $\eta = \eta'/\rho_{2s+r}^+$ such that $\kappa - \frac{1}{\sqrt{2}+\nu} < \eta' \leq 1$. Then PHT($r$) recovers the support of $\bar{\boldsymbol{x}}$ within*

$$t_{\max} = \left( \frac{\log \kappa}{\log(1/\beta)} + \frac{\log(\sqrt{2}/(1-\lambda))}{\log(1/\beta)} + 2 \right) \|\bar{\boldsymbol{x}}\|_0$$

*iterations, provided that for some constant $\lambda \in (0, 1)$*

$$\bar{\boldsymbol{x}}_{\min} \geq \frac{2\nu + 6}{\lambda \rho_{2s+r}^-} \|\nabla_{s+r} F(\bar{\boldsymbol{x}})\|.$$

*Above, $\beta = (\sqrt{2} + \nu)(\kappa - \eta') \in (0, 1)$.*

We remark several aspects of the theorem. The most important part is that Theorem 3 offers the theoretical justification that PHT($r$) always recovers the support. This is achieved by imposing an RIP condition (i.e., bounding the condition number from the above) and using a proper step size.

We also make the iteration bound explicit, in order to examine the parameter dependency. First, we note that $t_{\max}$ scales approximately linearly with $\lambda$. This conforms the intuition because a small $\lambda$ actually indicates a large signal-to-noise ratio, and hence easy to distinguish the support of interest from the noise. The freedom parameter $r$ is mainly encoded in the coefficient $\beta$ through the quantity $\nu$. Observe that when increasing the scalar $r$, we have a small $\beta$, and hence fewer iterations. This is not surprising since a large value of $r$ grants the algorithm more freedom to look at the current iterate. Indeed, in the best case, PHT($s$) is able to recover the support in $O(1)$ iterations while PHT(1) has to take $O(s)$ steps. However, if we investigate the conditions, we find that we need a stronger RSC/RSS condition to afford a large freedom parameter.

It is also interesting to contrast Theorem 3 to [31, 4], which independently built state-of-the-art support recovery results for HTP. As has been mentioned, [31] made use of the optimality of the target signal, which is a restricted setting compared to our result. Their iteration bound (see Theorem 1 therein), though provides an appealing insight, does not have a clear parameter dependence on the natural parameters of the problem (e.g., sparsity and condition number). [4] developed $O(\|\bar{x}\|_0)$ iteration complexity for compressed sensing. Again, they confined to a special signal whereas we carry out a generalization that allows us to analyze a family of algorithms.

Though the RIP condition has been ubiquitous in the literature, many researchers point out that it is not realistic in practical applications [18, 20, 21]. This is true for large-scale machine learning problems, where the condition number may grow with the sample size (hence one cannot upper bound it with a constant). A clever solution was first (to our knowledge) suggested by [14], where they showed that using the sparsity parameter $k = O(\kappa^2 s)$ guarantees convergence of projected gradient descent. The idea was recently employed by [22, 31] to show an RIP-free condition for sparse recovery, though in a technically different way. The following theorem borrows this elegant idea to prove RIP-free results for PHT($r$).

**Theorem 4.** *Consider the PHT($r$) algorithm. Suppose that $F(x)$ is $\rho_{2k}^-$-RSC and $\rho_{2k}^+$-RSS. Let $\kappa := \rho_{2k}^+/\rho_{2k}^-$ be the condition number. Further pick $k \geq s + \left(1 + \frac{4}{\eta^2 (\rho_{2k}^-)^2}\right) \min\{s, r\}$ where $\eta \in (0, 1/\rho_{2k}^+)$. Then the support of $\bar{x}$ is included in the iterate of PHT($r$) within*

$$t_{\max} = \left(\frac{3 \log \kappa}{\log(1/\mu)} + \frac{2 \log(2/(1-\lambda))}{\log(1/\mu)} + 2\right) \|\bar{x}\|_0$$

*iterations, provided that for some constant $\lambda \in (0, 1)$,*

$$\bar{x}_{\min} \geq \frac{\sqrt{\kappa} + 3}{\lambda \rho_{2k}^-} \|\nabla_{k+s} F(\bar{x})\|.$$

*Above, we have $\mu = 1 - \frac{\eta \rho_{2k}^-(1 - \eta \rho_{2k}^+)}{2}$.*

We discuss the salient features of Theorem 4 compared to Prop. 2 and Theorem 3. First, note that we can pick $\eta = 1/(2\rho_{2k}^+)$ in the above theorem, which results in $\mu = O(1 - 1/\kappa)$. So the iteration complexity is essentially given by $O(s\kappa \log \kappa)$ that is similar to the one in Prop. 2. However, in Theorem 4, the sparsity parameter $k$ is set to be $O(s + \kappa^2 \min\{s, r\})$ which guarantees support inclusion. We pose an open question of whether the $\bar{x}_{\min}$-condition might be refined, in that it currently scales with $\sqrt{\kappa}$ which is stringent for ill-conditioned problems. Another important consequence implied by the theorem is that the sparsity parameter $k$ actually depends on the minimum of $s$ and $r$. Consider $r = 1$ which corresponds to the OMPR algorithm. Then $k = O(s + \kappa^2)$ suffices. In contrast, previous work of [14, 31, 22, 23] only obtained theoretical result for $k = O(\kappa^2 s)$, owing to a restricted problem setting. We also note that even in the original OMPR paper [12] and its latest version [13], such an RIP-free condition was not established.

## 2.2 Statistical Results

Until now, all of our theoretical results are phrased in terms of deterministic conditions (i.e., RSC, RSS and $\bar{x}_{\min}$). It is known that these conditions can be satisfied by prevalent statistical models

such as linear regression and logistic regression. Here, we give detailed statistical results for sparse linear regression, and we refer the reader to [1, 14, 22, 23] for other applications.

Consider the sparse linear regression model

$$y_i = \langle \boldsymbol{a}_i, \bar{\boldsymbol{x}} \rangle + e_i, \quad 1 \le i \le N,$$

where $\boldsymbol{a}_i$ are drawn i.i.d. from a sub-gaussian distribution with zero mean and covariance $\boldsymbol{\Sigma} \in \mathbb{R}^{d \times d}$ and $e_i$ are drawn i.i.d. from $\mathcal{N}(0, \omega^2)$. We presume that the diagonal elements of $\boldsymbol{\Sigma}$ are properly scaled, i.e., $\Sigma_{jj} \le 1$ for $1 \le j \le d$. Let $\boldsymbol{A} = (\boldsymbol{a}_1^\top; \ldots; \boldsymbol{a}_N^\top)$ and $\boldsymbol{y} = (y_1; \ldots; y_N)$. Our goal is to recover $\bar{\boldsymbol{x}}$ from the knowledge of $\boldsymbol{A}$ and $\boldsymbol{y}$. To this end, we may choose $F(\boldsymbol{x}) = \frac{1}{2} \|\boldsymbol{y} - \boldsymbol{A}\boldsymbol{x}\|^2$. Let $\sigma_{\min}(\boldsymbol{\Sigma})$ and $\sigma_{\max}(\boldsymbol{\Sigma})$ be the smallest and the largest singulars of $\boldsymbol{\Sigma}$, respectively. Then it is known that with high probability, $F(\boldsymbol{x})$ satisfies the RSC and RSS properties at the sparsity level $K$ with parameters

$$\rho_K^- = \sigma_{\min}(\boldsymbol{\Sigma}) - \mathrm{C}_1 \cdot \frac{K \log d}{N}, \quad \rho_K^+ = \sigma_{\max}(\boldsymbol{\Sigma}) + \mathrm{C}_2 \cdot \frac{K \log d}{N}, \tag{3}$$

respectively. It is also known that with high probability, the following holds:

$$\|\nabla_K F(\bar{\boldsymbol{x}})\| \le 2\omega \sqrt{\frac{K \log d}{N}}. \tag{4}$$

See [1] for a detailed discussion. With these probabilistic arguments on hand, we investigate the sufficient conditions under which the preceding deterministic results hold.

For Prop. 2, recall that the sparsity level of RSC and RSS is $2s$. Hence, if we pick the sample size $N = q \cdot 2\mathrm{C}_1 s \log d / \sigma_{\min}(\boldsymbol{\Sigma})$ for some $q > 1$, then

$$\frac{4\sqrt{2} + 2\sqrt{\kappa_{2s}}}{\rho_{2s}^-} \|\nabla_{2s} F(\bar{\boldsymbol{x}})\| \le 4\omega \frac{2\sqrt{2} + \sqrt{\frac{\sigma_{\max}(\boldsymbol{\Sigma})}{\sigma_{\min}(\boldsymbol{\Sigma})}} \cdot \sqrt{\frac{1 + \mathrm{C}_2 / q\mathrm{C}_1}{1 - 1/q}}}{(1 - 1/q)\sqrt{q\mathrm{C}_1 \sigma_{\min}(\boldsymbol{\Sigma})}}.$$

The right-hand side is monotonically decreasing with $q$, which indicates that as soon as we pick $q$ large enough, it becomes smaller than $\bar{\boldsymbol{x}}_{\min}$. To be more concrete, consider that the covariance matrix $\boldsymbol{\Sigma}$ is the identity matrix for which $\sigma_{\min}(\boldsymbol{\Sigma}) = \sigma_{\max}(\boldsymbol{\Sigma}) = 1$. Now suppose that $q \ge 2$, which gives an upper bound

$$\frac{4\sqrt{2} + 2\sqrt{\kappa_{2s}}}{\rho_{2s}^-} \|\nabla_{2s} F(\bar{\boldsymbol{x}})\| \le \frac{8\omega(2\sqrt{2} + \sqrt{2 + \mathrm{C}_2/\mathrm{C}_1})}{\sqrt{q\mathrm{C}_1}}.$$

Thus, in order to fulfill the $\bar{\boldsymbol{x}}_{\min}$-condition in Prop. 2, it suffices to pick

$$q = \max \left\{ 2, \left( \frac{8\omega(2\sqrt{2} + \sqrt{2 + \mathrm{C}_2/\mathrm{C}_1})}{\sqrt{\mathrm{C}_1}\bar{\boldsymbol{x}}_{\min}} \right)^2 \right\}.$$

For Theorem 3, it essentially asks for a well-conditioned design matrix at the sparsity level $2s + r$. Note that (3) implies $\kappa_{2s+r} \ge \sigma_{\max}(\boldsymbol{\Sigma})/\sigma_{\min}(\boldsymbol{\Sigma})$, which in return requires a well-conditioned covariance matrix. Thus, to guarantee that $\kappa_{2s+r} \le 1 + \epsilon$ for some $\epsilon > 0$, it suffices to choose $\boldsymbol{\Sigma}$ such that $\sigma_{\max}(\boldsymbol{\Sigma})/\sigma_{\min}(\boldsymbol{\Sigma}) < 1 + \epsilon$ and pick $N = q \cdot \mathrm{C}_1(2s + r) \log d / \sigma_{\min}(\boldsymbol{\Sigma})$ with

$$q = \frac{1 + \epsilon + \mathrm{C}_1^{-1}\mathrm{C}_2 \sigma_{\max}(\boldsymbol{\Sigma})/\sigma_{\min}(\boldsymbol{\Sigma})}{1 + \epsilon - \sigma_{\max}(\boldsymbol{\Sigma})/\sigma_{\min}(\boldsymbol{\Sigma})}.$$

Finally, Theorem 4 asserts support inclusion by expanding the support size of the iterates. Suppose that $\eta = 1/(2\rho_{2k}^+)$, which results in $k \ge s + (16\kappa_{2k}^2 + 1) \min\{r, s\}$. Given that the condition number $\kappa_{2k}$ is always greater than 1, we can pick $k \ge s + 20\kappa_{2k}^2 \min\{r, s\}$. At a first sight, this seems to be weird in that $k$ depends on the condition number $\kappa_{2k}$ which itself relies on the choice of $k$. In the following, we present concrete sample complexity showing that this condition can be met. We will focus on two extreme cases: $r = 1$ and $r = s$.

For $r = 1$, we require $k \ge s + 20\kappa_{2k}^2$. Let us pick $N = 4\mathrm{C}_1 k \log d / \sigma_{\min}(\boldsymbol{\Sigma})$. In this way, we obtain $\rho_{2k}^- = \frac{1}{2}\sigma_{\min}(\boldsymbol{\Sigma})$ and $\rho_{2k}^+ \le (1 + \frac{\mathrm{C}_2}{2\mathrm{C}_1})\sigma_{\max}(\boldsymbol{\Sigma})$. It then follows that the condition number of the design matrix $\kappa_{2k} \le (2 + \frac{\mathrm{C}_2}{\mathrm{C}_1})\sigma_{\max}(\boldsymbol{\Sigma})/\sigma_{\min}(\boldsymbol{\Sigma})$. Consequently, we can set the parameter

$$k = s + 20 \left( \left( 2 + \frac{\mathrm{C}_2}{\mathrm{C}_1} \right) \frac{\sigma_{\max}(\boldsymbol{\Sigma})}{\sigma_{\min}(\boldsymbol{\Sigma})} \right)^2.$$

Note that the above quantities depend only on the covariance matrix. Again, if $\mathbf{\Sigma}$ is the identity matrix, the sample complexity is $O(s \log d)$.

For $r = s$, likewise $k \geq 20 \kappa_{2k}^2 s$ suffices. Following the deduction above, we get

$$k = 20 \left( \left( 2 + \frac{\mathrm{C}_2}{\mathrm{C}_1} \right) \frac{\sigma_{\max}(\mathbf{\Sigma})}{\sigma_{\min}(\mathbf{\Sigma})} \right)^2 s.$$

## 3   Proof Sketch

We sketch the proof and list some useful lemmas which might be of independent interest. The high-level proof technique follows from the recent work of [4] which performs an RIP analysis for compressed sensing. But for our purpose, we have to deal with the freedom parameter $r$ as well as the RIP-free condition. We also need to generalize the arguments in [4] to show support recovery results for arbitrary sparse signals. Indeed, we prove the following lemma which is crucial for our analysis. Below we assume without loss of generality that the elements in $\bar{\boldsymbol{x}}$ are in descending order according to the magnitude.

**Lemma 5.** *Consider the PHT(r) algorithm. Assume that $F(\boldsymbol{x})$ is $\rho_{2k}^-$-RSC and $\rho_{2k}^+$-RSS. Further assume that the sequence of $\{\boldsymbol{x}^t\}_{t \geq 0}$ satisfies*

$$\left\| \boldsymbol{x}^t - \bar{\boldsymbol{x}} \right\| \leq \alpha \cdot \beta^t \left\| \boldsymbol{x}^0 - \bar{\boldsymbol{x}} \right\| + \psi_1, \tag{5}$$

$$\left\| \boldsymbol{x}^t - \bar{\boldsymbol{x}} \right\| \leq \gamma \left\| \bar{\boldsymbol{x}}_{\overline{S^t}} \right\| + \psi_2, \tag{6}$$

*for positive $\alpha$, $\psi_1$, $\gamma$, $\psi_2$ and $0 < \beta < 1$. Suppose that at the $n$-th iteration ($n \geq 0$), $S^n$ contains the indices of top $p$ (in magnitude) elements of $\bar{\boldsymbol{x}}$. Then, for any integer $1 \leq q \leq s - p$, there exists an integer $\Delta \geq 1$ determined by*

$$\sqrt{2} \left| \bar{x}_{p+q} \right| > \alpha \gamma \cdot \beta^{\Delta - 1} \left\| \bar{\boldsymbol{x}}_{\{p+1,\dots,s\}} \right\| + \Psi,$$

*where*

$$\Psi = \alpha \psi_2 + \psi_1 + \frac{1}{\rho_{2k}} \left\| \nabla_2 F(\bar{\boldsymbol{x}}) \right\|,$$

*such that $S^{n+\Delta}$ contains the indices of top $p + q$ elements of $\bar{\boldsymbol{x}}$ provided that $\Psi \leq \sqrt{2} \lambda \bar{x}_{\min}$ for some constant $\lambda \in (0, 1)$.*

We isolate this lemma here since we find it inspiring and general. The lemma states that under proper conditions, as far as one can show that the sequence satisfies (5) and (6), then after a few iterations, PHT(r) captures more correct indices in the iterate. Note that the condition (5) states that the sequence should contract with a geometric rate, and the condition (6) follows immediately from the fully corrective step (i.e., minimizing $F(\boldsymbol{x})$ over the new support set).

The next theorem concludes that under the conditions of Lemma 5, the total iteration complexity for support recovery is proportional to the sparsity of the underlying signal.

**Theorem 6.** *Assume same conditions as in Lemma 5. Then PHT(r) successfully identifies the support of $\bar{\boldsymbol{x}}$ using $\left( \frac{\log 2}{2 \log(1/\beta)} + \frac{\log(\alpha \gamma / (1 - \lambda))}{\log(1/\beta)} + 2 \right) \left\| \bar{\boldsymbol{x}} \right\|_0$ number of iterations.*

The detailed proofs of these two results are given in the appendix. Armed with them, it remains to show that PHT(r) satisfies the condition (5) under different settings.

*Proof Sketch for Prop. 2.* We start with comparing $F(\boldsymbol{z}_{S^t}^t)$ and $F(\boldsymbol{x}^{t-1})$. For the sake, we record several important properties. First, due to the fully corrective step, the support set of $\nabla F(\boldsymbol{x}^{t-1})$ is orthogonal to $S^{t-1}$. That means for any subset $\Omega \subset S^{t-1}$, $\boldsymbol{z}_\Omega^t = \boldsymbol{x}_\Omega^{t-1}$ and for any set $\Omega \subset \overline{S^{t-1}}$, $\boldsymbol{z}_\Omega^t = -\eta \nabla_\Omega F(\boldsymbol{x}^{t-1})$. We also note that due to the PHT operator, any element of $\boldsymbol{z}_{S^t \backslash S^{t-1}}^t$ is not smaller than that of $\boldsymbol{z}_{S^{t-1} \backslash S^t}^t$. These critical facts together with the RSS condition result in

$$F(\boldsymbol{x}^t) - F(\boldsymbol{x}^{t-1}) \leq F(\boldsymbol{z}_{S^t}^t) - F(\boldsymbol{x}^{t-1}) \leq -\eta(1 - \eta \rho_{2s}^+) \left\| \left( \nabla F(\boldsymbol{x}^{t-1}) \right)_{S^t \backslash S^{t-1}} \right\|^2.$$

Since $S^t \setminus S^{t-1}$ consists of the top elements of $\nabla F(\boldsymbol{x}^{t-1})$, we can show that

$$\left\| \left( \nabla F(\boldsymbol{x}^{t-1}) \right)_{S^t \setminus S^{t-1}} \right\|^2 \geq \frac{2\rho_{2s}^- \left| S^t \setminus S^{t-1} \right|}{\left| S^t \setminus S^{t-1} \right| + \left| S \setminus S^{t-1} \right|} \left( F(\boldsymbol{x}^{t-1}) - F(\bar{\boldsymbol{x}}) \right).$$

Using the argument of Prop. 21, we establish the linear convergence of the iterates, i.e., the condition (5). The result then follows. $\square$

*Proof Sketch for Theorem 3.* To prove this theorem, we present a more careful analysis on the problem structure. In particular, let $T = \text{supp}\left( \nabla F(\boldsymbol{x}^{t-1}, r) \right)$, $J^t = S^{t-1} \cup T$, and consider the elements of $\nabla F(\boldsymbol{x}^{t-1})$. Since $T$ contains the largest elements, any element outside $T$ is smaller than those of $T$. Then we may compare the elements of $\nabla F(\boldsymbol{x}^{t-1})$ on $S \setminus T$ and $S \setminus T$. Though they have different number of components, we can show the relationship between the averaged energy:

$$\frac{1}{|T \setminus S|} \left\| \left( \nabla F(\boldsymbol{x}^{t-1}) \right)_{T \setminus S} \right\|^2 \geq \frac{1}{|S \setminus T|} \left\| \left( \nabla F(\boldsymbol{x}^{t-1}) \right)_{S \setminus T} \right\|^2.$$

Using this equation followed by some standard relaxation, we can bound $\left\| \bar{\boldsymbol{x}}_{\overline{J^t}} \right\|$ in terms of $\left\| \boldsymbol{x}^{t-1} - \bar{\boldsymbol{x}} \right\|$ as follows.

**Lemma 7.** *Assume that $F(\boldsymbol{x})$ satisfies the properties of RSC and RSS at sparsity level $k + s + r$. Let $\rho^- := \rho_{k+s+r}^-$ and $\rho^+ := \rho_{k+s+r}^+$. Consider the support set $J^t = S^{t-1} \cup \text{supp}\left( \nabla F(\boldsymbol{x}^{t-1}), r \right)$. We have for any $0 < \theta \leq 1/\rho^+$,*

$$\left\| \bar{\boldsymbol{x}}_{\overline{J^t}} \right\| \leq \nu(1 - \theta\rho^-) \left\| \boldsymbol{x}^{t-1} - \bar{\boldsymbol{x}} \right\| + \frac{\nu}{\rho^-} \left\| \nabla_{s+r} F(\bar{\boldsymbol{x}}) \right\|,$$

*where $\nu = \sqrt{s - r + 2}$. In particular, picking $\theta = 1/\rho^+$ gives*

$$\left\| \bar{\boldsymbol{x}}_{\overline{J^t}} \right\| \leq \nu \left( 1 - \frac{1}{\kappa} \right) \left\| \boldsymbol{x}^{t-1} - \bar{\boldsymbol{x}} \right\| + \frac{\nu}{\rho^-} \left\| \nabla_{s+r} F(\bar{\boldsymbol{x}}) \right\|.$$

Note that the lemma also applies to the two-stage thresholding algorithms (e.g., CoSaMP [15]) whose first step is expanding the support set.

On the other hand, we also know that

$$\left\| \boldsymbol{z}_{J^t \setminus S^t}^t \right\| \leq \left\| \boldsymbol{z}_{J^t \setminus S}^t \right\|.$$

This is because $J^t \setminus S^t$ contains the $r$ smallest elements of $\boldsymbol{z}_{J^t}^t$. It then follows that $\left\| \bar{\boldsymbol{x}}_{J^t \setminus S^t} \right\|$ can be upper bounded by $\left\| \boldsymbol{x}^{t-1} - \bar{\boldsymbol{x}} \right\|$. Finally, we note that $\overline{S^t} = (J^t \setminus S^t) \cup \overline{J^t}$. Hence, (5) follows. $\square$

*Proof Sketch for Theorem 4.* The proof idea of Theorem 4 is inspired by [31], though we give a tighter and a more general analysis. We first observe that $S^t \setminus S^{t-1}$ contains larger elements than $S^{t-1} \setminus S^t$, due to PHT. This enables us to show that

$$F(\boldsymbol{x}^t) - F(\boldsymbol{x}^{t-1}) \leq -\frac{1 - \eta\rho_{2k}^+}{2\eta} \left\| \boldsymbol{z}_{S^t}^t - \boldsymbol{x}^{t-1} \right\|^2 \leq -\frac{1 - \eta\rho_{2k}^+}{2\eta} \left\| \left( \nabla F(\boldsymbol{x}^{t-1}) \right)_{S^t \setminus S^{t-1}} \right\|^2.$$

Then we prove the claim

$$\left\| \left( \nabla F(\boldsymbol{x}^{t-1}) \right)_{S^t \setminus S^{t-1}} \right\|^2 \geq \rho_{2k}^- \left( F(\boldsymbol{x}^{t-1}) - F(\bar{\boldsymbol{x}}) \right).$$

To this end, we consider whether $r$ is larger than $s$. If $r \geq s$, then it is possible that $\left| S^t \setminus S^{t-1} \right| \geq s$. In this case, using the RSC condition and the PHT property, we can show that

$$\left\| \left( \nabla F(\boldsymbol{x}^{t-1}) \right)_{S^t \setminus S^{t-1}} \right\|^2 \geq \left\| \left( \nabla F(\boldsymbol{x}^{t-1}) \right)_{S \setminus S^{t-1}} \right\|^2 \geq \rho_{2k}^- \left( F(\boldsymbol{x}^{t-1}) - F(\bar{\boldsymbol{x}}) \right).$$

If $\left| S^t \setminus S^{t-1} \right| < s \leq r$, then the above does not hold. But we may partition the set $S \setminus S^{t-1}$ as a union of $T_1 = S \setminus (S^t \cup S^{t-1})$ and $T_2 = (S^t \setminus S^{t-1}) \cap S$, and show that the $\ell_2$-norm of $F(\boldsymbol{x}^{t-1})$ on $T_1$ is smaller than that on $T_2$ if $k = s + \kappa^2 s$. In addition, the RSC condition gives

$$\frac{\rho_{2k}^-}{4} \left\| \bar{\boldsymbol{x}} - \boldsymbol{x}^{t-1} \right\|^2 \leq F(\bar{\boldsymbol{x}}) - F(\boldsymbol{x}^{t-1}) + \frac{1}{\rho_{2k}^-} \left\| \left( \nabla F(\boldsymbol{x}^{t-1}) \right)_{T_1} \right\|^2 + \frac{1}{\rho_{2k}^-} \left\| \left( \nabla F(\boldsymbol{x}^{t-1}) \right)_{S^t \setminus S^{t-1}} \right\|^2.$$

Since $T_2 \subset S^t \setminus S^{t-1}$, it implies the desired bound by rearranging the terms.

The case $r < s$ follows in a reminiscent way. The proof is complete. $\square$

# 4  Simulation

We complement our theoretical results by performing numerical experiments in this section. In particular, we are interested in two aspects: first, the number of iterations required to identify the support of an $s$-sparse signal; second, the tradeoff between the iteration number and percentage of success resulted from different choices of the freedom parameter $r$.

We consider the compressed sensing model $\boldsymbol{y} = \boldsymbol{A}\bar{\boldsymbol{x}} + 0.01\boldsymbol{e}$, where the dimension $d = 200$ and the entries of $\boldsymbol{A}$ and $\boldsymbol{e}$ are i.i.d. normal variables. Given a sparsity level $s$, we first uniformly choose the support of $\bar{\boldsymbol{x}}$, and assign values to the non-zeros with i.i.d. normals. There are two configurations: the sparsity $s$ and the sample size $N$. Given $s$ and $N$, we independently generate 100 signals and test PHT$(r)$ on them. We say PHT$(r)$ succeeds in a trial if it returns an iterate with correct support within 10 thousands iterations. Otherwise we mark the trial as failure. Iteration numbers to be reported are counted only on those success trials. The step size $\eta$ is fixed to be the unit, though one can tune it using cross-validation for better performance.

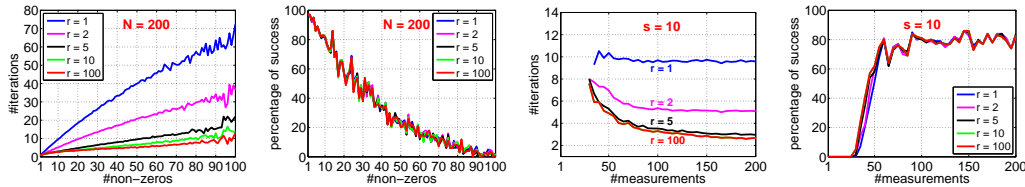

Figure 1: **Iteration number and success percentage against sparsity and sample size.** The first panel shows that the iteration number grows linearly with the sparsity. The choice $r = 5$ suffices to guarantee a minimum iteration complexity. The second panel shows comparable statistical performance for different choices of $r$. The third one illustrates how the iteration number changes with the sample size and the last panel depicts phase transition.

To study how the iteration number scales with the sparsity in practice, we fix $N = 200$ and tune $s$ from 1 to 100. We test different freedom parameter $r$ on these signals. The results are shown in the leftmost figure in Figure 1. As our theory predicted, we observe that within $O(s)$ iterations, PHT$(r)$ precisely identifies the true support. In the second subfigure, we plot the percentage of success against the sparsity. It appears that PHT$(r)$ lays on top of each other. This is possibly because we used a sufficiently large sample size.

Next, we fix $s = 10$ and vary $N$ from 1 to 200. Surprisingly, from the rightmost figure, we do not observe performance degrade using a large freedom parameter. So we conjecture that the $\bar{\boldsymbol{x}}_{\min}$-condition we established can be refined.

Figure 1 also illustrates an interesting phenomenon: after a particular threshold, say $r = 5$, PHT$(r)$ does not significantly reduces the iteration number by increasing $r$. This cannot be explained by our theorems in the paper. We leave it as a promising research direction.

# 5  Conclusion and Future Work

In this paper, we have presented a principled analysis on a family of hard thresholding algorithms. To facilitate our analysis, we appealed to the recently proposed partial hard thresholding operator. We have shown that under the RIP condition or the relaxed sparsity condition, the PHT$(r)$ algorithm recovers the support of an arbitrary sparse signal $\bar{\boldsymbol{x}}$ within $O(\|\bar{\boldsymbol{x}}\|_0 \kappa \log \kappa)$ iterations, provided that a generalized signal-to-noise ratio condition is satisfied. On account of our unified analysis, we have established the best known bound for HTP and OMPR. We have also illustrated that the simulation results agree with our finding that the iteration number is proportional to the sparsity.

There are several interesting future directions. First, it would be interesting to examine if we can close the logarithmic factor $\log \kappa$ in the iteration bound. Second, it is also useful to study RIP-free conditions for two-stage PHT algorithms such as CoSaMP. Finally, we pose the open question of whether one can improve the $\sqrt{\kappa}$ factor in the $\bar{\boldsymbol{x}}_{\min}$-condition.

**Acknowledgements.** The work is supported in part by NSF-Bigdata-1419210 and NSF-III-1360971. We thank the anonymous reviewers for valuable comments.

## Footnotes

[1]Our results actually hold for $|T| \leq k$. But we observe that the size of $T$ we will consider is usually equal to $k$. Hence, for ease of exposition, we take $|T| = k$. This is also the case considered in [12].

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
