[Supplementary Material · supp.pdf]

# A    Proof for Section 2

Throughout our proof, we presume without loss of generality that the elements in $\bar{\boldsymbol{x}} = (\bar{x}_1, \bar{x}_2, \ldots, \bar{x}_d)$ are in descending order by their magnitude, i.e., $|\bar{x}_1| \geq |\bar{x}_2| \geq \cdots \geq |\bar{x}_s|$ and $\bar{x}_i = 0$ for $s < i \leq d$. We also write $[n] := \{1, 2, \ldots, n\}$ for brevity.

Recall that the partial hard thresholding algorithm with freedom parameter $r$ proceeds as follows at the $t$-th iteration:

$$\boldsymbol{z}^t = \boldsymbol{x}^{t-1} - \eta \nabla F(\boldsymbol{x}^{t-1})$$
$$J^t = S^{t-1} \cup \operatorname{supp}\left(\nabla F(\boldsymbol{x}^{t-1}), r\right)$$
$$\boldsymbol{y}^t = \operatorname{HT}_k(\boldsymbol{z}^t_{J^t})$$
$$S^t = \operatorname{supp}\left(\boldsymbol{y}^t\right)$$
$$\boldsymbol{x}^t = \underset{\operatorname{supp}(\boldsymbol{x}) \subset S^t}{\arg\min}\ F(\boldsymbol{x})$$

We first prove the results that appear in Section 3.

**Lemma 8** (Restatement of Lemma 5). *Assume that $F(\boldsymbol{x})$ is $\rho_{2k}^-$-RSC and $\rho_{2k}^+$-RSS. Consider the PHT($r$) algorithm with $\eta < 1/\rho_{2k}^+$. Further assume that the sequence of $\{\boldsymbol{x}^t\}_{t \geq 0}$ satisfies*

$$\left\| \boldsymbol{x}^t - \bar{\boldsymbol{x}} \right\| \leq \alpha \cdot \beta^t \left\| \boldsymbol{x}^0 - \bar{\boldsymbol{x}} \right\| + \psi_1,$$
$$\left\| \boldsymbol{x}^t - \bar{\boldsymbol{x}} \right\| \leq \gamma \left\| \bar{\boldsymbol{x}}_{\overline{S^t}} \right\| + \psi_2,$$

*for positive $\alpha$, $\psi_1$, $\gamma$, $\psi_2$ and $0 < \beta < 1$. Suppose that at the $n$-th iteration ($n \geq 0$), $S^n$ contains the indices of top $p$ (in magnitude) elements of $\bar{\boldsymbol{x}}$. Then, for any integer $1 \leq q \leq s - p$, there exists an integer $\Delta \geq 1$ determined by*

$$\sqrt{2}\,|\bar{x}_{p+q}| > \alpha\gamma \cdot \beta^{\Delta-1} \left\| \bar{\boldsymbol{x}}_{\{p+1, \ldots, s\}} \right\| + \Psi$$

*where*

$$\Psi = \alpha\psi_2 + \psi_1 + \frac{1}{\rho_{2k}^-} \left\| \nabla_2 F(\bar{\boldsymbol{x}}) \right\|,$$

*such that $S^{n+\Delta}$ contains the indices of top $p + q$ elements of $\bar{\boldsymbol{x}}$ provided that $\Psi \leq \sqrt{2}\lambda\bar{x}_{\min}$ for some $\lambda \in (0, 1)$.*

*Proof.* We aim at deriving a condition under which $[p + q] \subset S^{n+\Delta}$. To this end, it suffices to enforce

$$\min_{j \in [p+q]} \left| z_j^{n+\Delta} \right| > \max_{i \in \overline{S}} \left| z_i^{n+\Delta} \right|. \tag{7}$$

On one hand, for any $j \in [p + q]$,

$$\left| z_j^{n+\Delta} \right| = \left| \left( \boldsymbol{x}^{n+\Delta-1} - \eta \nabla F(\boldsymbol{x}^{n+\Delta-1}) \right)_j \right|$$
$$\geq |\bar{x}_j| - \left| \left( \boldsymbol{x}^{n+\Delta-1} - \bar{\boldsymbol{x}} - \eta \nabla F(\boldsymbol{x}^{n+\Delta-1}) \right)_j \right|$$
$$\geq |\bar{x}_{p+q}| - \left| \left( \boldsymbol{x}^{n+\Delta-1} - \bar{\boldsymbol{x}} - \eta \nabla F(\boldsymbol{x}^{n+\Delta-1}) \right)_j \right|.$$

On the other hand, for all $i \in \overline{S}$,

$$\left| z_i^{n+\Delta} \right| = \left| \left( \boldsymbol{x}^{n+\Delta-1} - \bar{\boldsymbol{x}} - \eta \nabla F(\boldsymbol{x}^{n+\Delta-1}) \right)_i \right|.$$

Hence, we know that to guarantee (7), it suffices to ensure for all $j \in [p + q]$ and $i \in \overline{S}$ that

$$|\bar{x}_{p+q}| > \left| \left( \boldsymbol{x}^{n+\Delta-1} - \bar{\boldsymbol{x}} - \eta \nabla F(\boldsymbol{x}^{n+\Delta-1}) \right)_j \right| + \left| \left( \boldsymbol{x}^{n+\Delta-1} - \bar{\boldsymbol{x}} - \eta \nabla F(\boldsymbol{x}^{n+\Delta-1}) \right)_i \right|.$$

Note that the right-hand side is upper bounded as follows:

$$\frac{1}{\sqrt{2}}\left|\left(\boldsymbol{x}^{n+\Delta-1}-\bar{\boldsymbol{x}}-\eta\nabla F(\boldsymbol{x}^{n+\Delta-1})\right)_j\right| + \frac{1}{\sqrt{2}}\left|\left(\boldsymbol{x}^{n+\Delta-1}-\bar{\boldsymbol{x}}-\eta\nabla F(\boldsymbol{x}^{n+\Delta-1})\right)_i\right|$$

$$\leq \left\|\left(\boldsymbol{x}^{n+\Delta-1}-\bar{\boldsymbol{x}}-\eta\nabla F(\boldsymbol{x}^{n+\Delta-1})\right)_{\{j,i\}}\right\|$$

$$\leq \left\|\left(\boldsymbol{x}^{n+\Delta-1}-\bar{\boldsymbol{x}}-\eta\nabla F(\boldsymbol{x}^{n+\Delta-1})+\eta\nabla F(\bar{\boldsymbol{x}})\right)_{\{j,i\}}\right\| + \eta\left\|\left(\nabla F(\bar{\boldsymbol{x}})\right)_{\{j,i\}}\right\|$$

$$\leq \phi_{2k}\left\|\boldsymbol{x}^{n+\Delta-1}-\bar{\boldsymbol{x}}\right\| + \eta\left\|\nabla_2 F(\bar{\boldsymbol{x}})\right\|$$

$$\leq \phi_{2k}\alpha\cdot\beta^{\Delta-1}\left\|\boldsymbol{x}^n-\bar{\boldsymbol{x}}\right\| + \phi\psi_1 + \eta\left\|\nabla_2 F(\bar{\boldsymbol{x}})\right\|,$$

where $\phi_{2k}$ is given by Lemma 17. Note that $\phi_{2k}<1$ whenever $0<\eta<1/\rho_{2k}^+$. Moreover,

$$\left\|\boldsymbol{x}^n-\bar{\boldsymbol{x}}\right\| \leq \gamma\left\|\bar{\boldsymbol{x}}_{\overline{S^n}}\right\| + \psi_2 \leq \gamma\left\|\bar{\boldsymbol{x}}_{\overline{[p]}}\right\| + \psi_2 = \gamma\left\|\bar{\boldsymbol{x}}_{\{p+1,\ldots,s\}}\right\| + \psi_2.$$

Put all the pieces together, we have

$$\frac{1}{\sqrt{2}}\left|\left(\boldsymbol{x}^{n+\Delta-1}-\bar{\boldsymbol{x}}-\eta\nabla F(\boldsymbol{x}^{n+\Delta-1})\right)_j\right| + \frac{1}{\sqrt{2}}\left|\left(\boldsymbol{x}^{n+\Delta-1}-\bar{\boldsymbol{x}}-\eta\nabla F(\boldsymbol{x}^{n+\Delta-1})\right)_i\right|$$

$$\leq \alpha\gamma\cdot\beta^{\Delta-1}\left\|\bar{\boldsymbol{x}}_{\{p+1,\ldots,s\}}\right\| + \alpha\psi_2 + \psi_1 + \eta\left\|\nabla_2 F(\bar{\boldsymbol{x}})\right\|$$

$$\leq \alpha\gamma\cdot\beta^{\Delta-1}\left\|\bar{\boldsymbol{x}}_{\{p+1,\ldots,s\}}\right\| + \alpha\psi_2 + \psi_1 + \frac{1}{\rho_{2k}^-}\left\|\nabla_2 F(\bar{\boldsymbol{x}})\right\|.$$

Therefore, when

$$\sqrt{2}\left|\bar{x}_{p+q}\right| > \alpha\gamma\cdot\beta^{\Delta-1}\left\|\bar{\boldsymbol{x}}_{\{p+1,\ldots,s\}}\right\| + \alpha\psi_2 + \psi_1 + \frac{1}{\rho_{2k}^-}\left\|\nabla_2 F(\bar{\boldsymbol{x}})\right\|,$$

we always have (7). Note that the above holds as far as $\Psi := \alpha\psi_2 + \psi_1 + \frac{1}{\rho_{2k}^-}\left\|\nabla_2 F(\bar{\boldsymbol{x}})\right\|$ is strictly smaller than $\sqrt{2}\left|\bar{x}_s\right|$. $\qquad\square$

**Theorem 9** (Restatement of Theorem 6). *Assume same conditions as in Lemma 5. Then PHT$(r)$ successfully identifies the support of $\bar{\boldsymbol{x}}$ using $\left(\frac{\log 2}{2\log(1/\beta)} + \frac{\log(\alpha\gamma/(1-\lambda))}{\log(1/\beta)} + 2\right)s$ number of iterations.*

*Proof.* We partition the support set $S=[s]$ into $K$ folds $S_1, S_2, \ldots, S_K$, where each $S_i$ is defined as follows:

$$S_i = \{s_{i-1}+1, \ldots, s_i\}, \ \forall\, 1\leq i\leq K.$$

Here, $s_0=0$ and for all $1\leq i\leq K$, the quantity $s_i$ is inductively given by

$$s_i = \max\left\{q:\ s_{i-1}+1\leq q\leq s \text{ and } |\bar{x}_q| > \frac{1}{\sqrt{2}}\left|\bar{x}_{s_{i-1}+1}\right|\right\}.$$

In this way, we note that for any two index sets $S_i$ and $S_j$, $S_i\cap S_j=\emptyset$ if $i\neq j$. We also know by the definition of $s_i$ that

$$\left|\bar{x}_{s_i+1}\right| \leq \frac{1}{\sqrt{2}}\left|\bar{x}_{s_{i-1}+1}\right|, \ \forall\, 1\leq i\leq K-1. \tag{8}$$

Now we show that after a finite number of iterations, say $n$, the union of the $S_i$'s is contained in $S^n$, i.e., the support set of the iterate $\boldsymbol{x}^n$. To this end, we prove that for all $0\leq i\leq K$,

$$\bigcup_{t=0}^{i} S_t \subset S^{n_0+n_1+\cdots+n_i} \tag{9}$$

for some $n_i$'s given below. Above, $S_0=\emptyset$.

We pick $n_0=0$ and it is easy to verify that $S_0\subset S^0$. Now suppose that (9) holds for $i-1$. That is, the index set of the top $s_{i-1}$ elements of $\bar{\boldsymbol{x}}$ is contained in $S^{n_0+\cdots+n_{i-1}}$. Due to Lemma 5, (9) holds for $i$ as long as $n_i$ satisfies

$$\sqrt{2}\left|\bar{x}_{s_i}\right| > \alpha\gamma\cdot\beta^{n_i-1}\left\|\bar{\boldsymbol{x}}_{\{s_{i-1}+1,\ldots,s\}}\right\| + \Psi, \tag{10}$$

where $\Psi$ is given in Lemma 5. Note that

$$
\begin{aligned}
\left\|\bar{\boldsymbol{x}}_{\{s_{i-1}+1,\ldots,s\}}\right\|^2 &= \left\|\bar{\boldsymbol{x}}_{S_i}\right\|^2 + \cdots + \left\|\bar{\boldsymbol{x}}_{S_K}\right\|^2 \\
&\leq (\bar{x}_{s_{i-1}+1})^2 |S_i| + \cdots + (\bar{x}_{s_{r-1}+1})^2 |S_K| \\
&\leq (\bar{x}_{s_{i-1}+1})^2 \left(|S_i| + 2^{-1} |S_{i+1}| + \cdots + 2^{i-K} |S_K|\right) \\
&< 2(\bar{x}_{s_i})^2 \left(|S_i| + 2^{-1} |S_{i+1}| + \cdots + 2^{i-K} |S_K|\right),
\end{aligned}
$$

where the second inequality follows from (8) and the last inequality follows from the definition of $q_i$. Denote for simplicity

$$
W_i := |S_i| + 2^{-1} |S_{i+1}| + \cdots + 2^{i-K} |S_K|.
$$

As we assumed $\Psi \leq \sqrt{2}\lambda \bar{\boldsymbol{x}}_{\min}$, we get

$$
\alpha\gamma \cdot \beta^{n_i-1} \left\|\bar{\boldsymbol{x}}_{\{s_{i-1}+1,\ldots,s\}}\right\| + \Psi < \sqrt{2}\alpha\gamma |\bar{x}_{s_i}| \beta^{n_i-1} \sqrt{W_i} + \sqrt{2}\lambda |\bar{x}_{s_i}|.
$$

Picking

$$
n_i = \log_{1/\beta} \frac{\alpha\gamma\sqrt{W_i}}{1-\lambda} + 2
$$

guarantees (10). It remains to calculate the total number of iterations. In fact, we have

$$
\begin{aligned}
t_{\max} &= n_0 + n_1 + \cdots + n_K \\
&= \frac{1}{2\log(1/\beta)} \sum_{i=1}^{K} \log W_i + K \cdot \frac{\log(\alpha\gamma/(1-\lambda))}{\log(1/\beta)} + 2K \\
&\overset{\zeta_1}{\leq} \frac{K}{2\log(1/\beta)} \log\left(\frac{1}{K}\sum_{i=1}^{K} W_i\right) + \left(\frac{\log(\alpha\gamma/(1-\lambda))}{\log(1/\beta)} + 2\right) K \\
&\overset{\zeta_2}{\leq} \frac{K}{2\log(1/\beta)} \log\left(\frac{2}{K}\sum_{i=1}^{K} |S_i|\right) + \left(\frac{\log(\alpha\gamma/(1-\lambda))}{\log(1/\beta)} + 2\right) K \\
&= \frac{K}{2\log(1/\beta)} \log\frac{2s}{K} + \left(\frac{\log(\alpha\gamma/(1-\lambda))}{\log(1/\beta)} + 2\right) K \\
&\overset{\zeta_3}{\leq} \left(\frac{\log 2}{2\log(1/\beta)} + \frac{\log(\alpha\gamma/(1-\lambda))}{\log(1/\beta)} + 2\right) s.
\end{aligned}
$$

Above, $\zeta_1$ immediately follows by observing that the logarithmic function is concave. $\zeta_2$ uses the fact that after rearrangement, the coefficient of $|S_i|$ is $\sum_{j=0}^{i-1} 2^{-j}$ which is always smaller than 2. Finally, since the function $a\log(2s/a)$ is monotonically increasing with respect to $a$ and $1 \leq a \leq s$, $\zeta_3$ follows. □

**Lemma 10** (Restatement of Lemma 7). *Assume that $F(\boldsymbol{x})$ satisfies the properties of RSC and RSS at sparsity level $k + s + r$. Let $\rho^- := \rho^-_{k+s+r}$ and $\rho^+ := \rho^+_{k+s+r}$. Consider the support set $J^t = S^{t-1} \cup \mathrm{supp}\left(\nabla F(\boldsymbol{x}^{t-1}), r\right)$. We have for any $0 < \theta \leq 1/\rho^+$,*

$$
\left\|\bar{\boldsymbol{x}}_{\overline{J^t}}\right\| \leq \nu(1 - \theta\rho^-) \left\|\boldsymbol{x}^{t-1} - \bar{\boldsymbol{x}}\right\| + \frac{\nu}{\rho^-} \left\|\nabla_{s+r} F(\bar{\boldsymbol{x}})\right\|,
$$

*where $\nu = \sqrt{s - r + 2}$. In particular, picking $\theta = 1/\rho^+$ gives*

$$
\left\|\bar{\boldsymbol{x}}_{\overline{J^t}}\right\| \leq \nu\left(1 - \frac{1}{\kappa}\right) \left\|\boldsymbol{x}^{t-1} - \bar{\boldsymbol{x}}\right\| + \frac{\nu}{\rho^-} \left\|\nabla_{s+r} F(\bar{\boldsymbol{x}})\right\|.
$$

*Proof.* Let $T = \mathrm{supp}\left(\nabla F(\boldsymbol{x}^{t-1}), r\right)$. Then $J^t = S^{t-1} \cup T$ and $S^{t-1} \cap T = \emptyset$. Since $T$ contains the top $r$ elements of $\nabla F(\boldsymbol{x}^{t-1})$, we have that each element in $T \setminus S$ is larger (in magnitude) than that in $S \setminus T$. In particular, we observe for $T \neq S$ that

$$
\frac{1}{|T \setminus S|} \left\|\left(\nabla F(\boldsymbol{x}^{t-1})\right)_{T \setminus S}\right\|^2 \geq \frac{1}{|S \setminus T|} \left\|\left(\nabla F(\boldsymbol{x}^{t-1})\right)_{S \setminus T}\right\|^2,
$$

which implies

$$\left\|\left(\nabla F(\boldsymbol{x}^{t-1})\right)_{T\setminus S}\right\| \geq \sqrt{\frac{r-|T\cap S|}{s-|T\cap S|}}\left\|\left(\nabla F(\boldsymbol{x}^{t-1})\right)_{S\setminus T}\right\| \geq \sqrt{\frac{1}{s-r+1}}\left\|\left(\nabla F(\boldsymbol{x}^{t-1})\right)_{S\setminus T}\right\|.$$

Since $\nabla F(\boldsymbol{x}^{t-1})$ is supported on $\overline{S^{t-1}}$, the LHS reads as

$$\left\|\left(\nabla F(\boldsymbol{x}^{t-1})\right)_{T\setminus S}\right\| = \left\|\left(\nabla F(\boldsymbol{x}^{t-1})\right)_{T\setminus(S\cup S^{t-1})}\right\| = \frac{1}{\theta}\left\|\left(\boldsymbol{x}^{t-1}-\theta\nabla F(\boldsymbol{x}^{t-1})-\bar{\boldsymbol{x}}\right)_{T\setminus(S\cup S^{t-1})}\right\|.$$

Now we look at the RHS. It follows that

$$\begin{aligned}
\left\|\left(\nabla F(\boldsymbol{x}^{t-1})\right)_{S\setminus T}\right\| &= \left\|\left(\nabla F(\boldsymbol{x}^{t-1})\right)_{S\setminus(T\cup S^{t-1})}\right\| \\
&= \frac{1}{\theta}\left\|\left(\boldsymbol{x}^{t-1}-\theta\nabla F(\boldsymbol{x}^{t-1})-\bar{\boldsymbol{x}}\right)_{S\setminus(T\cup S^{t-1})}+\bar{\boldsymbol{x}}_{S\setminus(T\cup S^{t-1})}\right\| \\
&\geq \frac{1}{\theta}\left\|\bar{\boldsymbol{x}}_{S\setminus(T\cup S^{t-1})}\right\|-\frac{1}{\theta}\left\|\left(\boldsymbol{x}^t-\theta\nabla F(\boldsymbol{x}^t)-\bar{\boldsymbol{x}}\right)_{S\setminus(T\cup S^{t-1})}\right\|.
\end{aligned}$$

Hence,

$$\begin{aligned}
&\left\|\bar{\boldsymbol{x}}_{\overline{J^t}}\right\| \\
&= \left\|\bar{\boldsymbol{x}}_{S\setminus(T\cup S^{t-1})}\right\| \\
&\leq \sqrt{s-r+1}\left\|\left(\boldsymbol{x}^{t-1}-\theta\nabla F(\boldsymbol{x}^{t-1})-\bar{\boldsymbol{x}}\right)_{T\setminus(S\cup S^{t-1})}\right\|+\left\|\left(\boldsymbol{x}^{t-1}-\theta\nabla F(\boldsymbol{x}^{t-1})-\bar{\boldsymbol{x}}\right)_{S\setminus(T\cup S^{t-1})}\right\| \\
&\leq \sqrt{s-r+1}\left\|\left(\boldsymbol{x}^{t-1}-\theta\nabla F(\boldsymbol{x}^{t-1})-\bar{\boldsymbol{x}}\right)_{T\setminus S}\right\|+\left\|\left(\boldsymbol{x}^{t-1}-\theta\nabla F(\boldsymbol{x}^{t-1})-\bar{\boldsymbol{x}}\right)_{S\setminus T}\right\| \\
&\leq \nu\left\|\left(\boldsymbol{x}^{t-1}-\theta\nabla F(\boldsymbol{x}^{t-1})-\bar{\boldsymbol{x}}\right)_{T\Delta S}\right\| \\
&\leq \nu\left\|\left(\boldsymbol{x}^{t-1}-\theta\nabla F(\boldsymbol{x}^{t-1})-\bar{\boldsymbol{x}}+\theta\nabla F(\bar{\boldsymbol{x}})\right)_{T\Delta S}\right\|+\nu\theta\left\|\left(\nabla F(\bar{\boldsymbol{x}})\right)_{T\Delta S}\right\| \\
&\leq \nu\phi_{k+s+r}\left\|\boldsymbol{x}^{t-1}-\bar{\boldsymbol{x}}\right\|+\nu\theta\left\|\left(\nabla F(\bar{\boldsymbol{x}})\right)_{T\Delta S}\right\|,
\end{aligned}$$

where $\nu=\sqrt{s-r+2}$ and the last inequality uses Lemma 18. For any $0<\theta\leq 1/\rho^+$, we have

$$\left\|\bar{\boldsymbol{x}}_{\overline{J^t}}\right\| \leq \nu(1-\theta m)\left\|\boldsymbol{x}^{t-1}-\bar{\boldsymbol{x}}\right\|+\frac{\nu}{\rho^-}\left\|\nabla_{s+r}F(\bar{\boldsymbol{x}})\right\|.$$

$\square$

## A.1 Proof of Prop. 2

*Proof.* Recall that we set $k=s$. Using Lemma 11, we have

$$F(\boldsymbol{x}^t)-F(\bar{\boldsymbol{x}}) \leq \mu_t\left(F(\boldsymbol{x}^{t-1})-F(\bar{\boldsymbol{x}})\right),$$

where $\mu_t=1-2\rho_{2s}^-\eta(1-\eta\rho_{2s}^+)\cdot\frac{|S^t\setminus S^{t-1}|}{|S^t\setminus S^{t-1}|+|S\setminus S^{t-1}|}$. Now combining this with Prop. 21, we have

$$\left\|\boldsymbol{x}^t-\bar{\boldsymbol{x}}\right\| \leq \sqrt{2\kappa}\sqrt{\mu_1\mu_2\ldots\mu_t}\left\|\boldsymbol{x}^0-\bar{\boldsymbol{x}}\right\|+\frac{3}{\rho_{2s}^-}\left\|\nabla_{2s}F(\bar{\boldsymbol{x}})\right\|.$$

Note that before the algorithm terminates, $1\leq\left|S^t\setminus S^{t-1}\right|\leq r$. Hence,

$$\mu_t \leq 1-\frac{2\eta\rho_{2s}^-(1-\eta\rho_{2s}^+)}{1+s} =: \mu.$$

It then follows that

$$\left\|\boldsymbol{x}^t-\bar{\boldsymbol{x}}\right\| \leq \sqrt{2\kappa}(\sqrt{\mu})^t\left\|\boldsymbol{x}^0-\bar{\boldsymbol{x}}\right\|+\frac{3}{\eta}\left\|\nabla_{2s}F(\bar{\boldsymbol{x}})\right\|. \tag{11}$$

Lemma 19 tells us

$$\left\|\boldsymbol{x}^t-\bar{\boldsymbol{x}}\right\| \leq \kappa\left\|\bar{\boldsymbol{x}}_{\overline{S^t}}\right\|+\frac{1}{\eta}\left\|\nabla_sF(\bar{\boldsymbol{x}})\right\|. \tag{12}$$

Hence, in light of Lemma 5 and Theorem 6, we obtain that PHT($r$) recovers the support using at most

$$t_{\max} = \left( \frac{\log 2}{\log(1/\mu)} + \frac{\log(2\kappa)}{\log(1/\mu)} + \frac{2\log(\kappa/(1-\lambda))}{\log(1/\mu)} + 2 \right) \|\bar{x}\|_0$$

iterations. Note that picking $\eta = O(1/\rho_{2s}^+)$, we have $\mu = O(1 - \frac{1}{\kappa})$ and $\log(1/\mu) = O(1/\kappa)$. This gives the $O(s\kappa \log \kappa)$ bound. $\quad\square$

**Lemma 11.** *Consider the PHT($r$) algorithm. Suppose that $F(x)$ is $\rho_{k+s}^-$-RSC and $\rho_{2k}^+$-RSS. Using the parameter $k = s$ and $\eta < 1/\rho_{2s}^+$, we have*

$$F(x^t) - F(\bar{x}) \leq \mu_t \left( F(x^{t-1}) - F(\bar{x}) \right),$$

*where $\mu_t = 1 - 2\eta\rho_{2s}^-(1 - \eta\rho_{2s}^+) \cdot \frac{|S^t \backslash S^{t-1}|}{|S^t\backslash S^{t-1}|+|S\backslash S^{t-1}|}$.*

*Proof.* Using the RSS property, we have

$$F(z_{S^t}^t) - F(x^{t-1}) \leq \left\langle \nabla F(x^{t-1}), z_{S^t}^t - x^{t-1} \right\rangle + \frac{\rho_{2s}^+}{2} \left\| z_{S^t}^t - x^{t-1} \right\|^2$$

$$\stackrel{\zeta_1}{=} \left\langle \nabla_{S^t\backslash S^{t-1}} F(x^{t-1}), z_{S^t\backslash S^{t-1}}^t \right\rangle + \frac{\rho_{2s}^+}{2} \left( \left\| z_{S^t\backslash S^{t-1}}^t \right\|^2 \right.$$

$$\left. + \left\| z_{S^t\cap S^{t-1}}^t - x_{S^t\cap S^{t-1}}^{t-1} \right\|^2 + \left\| x_{S^{t-1}\backslash S^t}^{t-1} \right\|^2 \right)$$

$$\stackrel{\zeta_2}{\leq} \left\langle \nabla_{S^t\backslash S^{t-1}} F(x^{t-1}), z_{S^t\backslash S^{t-1}}^t \right\rangle + \rho_{2s}^+ \left\| z_{S^t\backslash S^{t-1}}^t \right\|^2$$

$$\stackrel{\zeta_3}{=} - \eta(1 - \eta\rho_{2s}^+) \left\| \nabla_{S^t\backslash S^{t-1}} F(x^{t-1}) \right\|^2.$$

Above, we observe that $\nabla F(x^{t-1})$ is supported on $\overline{S^{t-1}}$ and we simply docompose the support set $S^t \cup S^{t-1}$ into three mutually disjoint sets, and hence $\zeta_1$ holds. To see why $\zeta_2$ holds, we note that for any set $\Omega \subset S^{t-1}$, $z_\Omega^t = x_\Omega^{t-1}$. Hence, $z_{S^t\cap S^{t-1}}^t = x_{S^t\cap S^{t-1}}^{t-1}$. Moreover, since $x_{S^{t-1}\backslash S^t}^{t-1} = z_{S^{t-1}\backslash S^t}^t$ and any element in $z_{S^{t-1}\backslash S^t}^t$ is not larger than that in $z_{S^t\backslash S^{t-1}}^t$ (recall that $S^t$ is obtained by hard thresholding), we have $\left\| x_{S^{t-1}\backslash S^t}^{t-1} \right\| \leq \left\| z_{S^t\backslash S^{t-1}}^t \right\|$ where we use the fact that $|S^t \backslash S^t| = |S^t \backslash S^{t-1}|$. Therefore, $\zeta_2$ holds. Finally, we write $z_{S^t\backslash S^{t-1}}^t = -\eta\nabla_{S^t\backslash S^{t-1}} F(x^{t-1})$ and obtain $\zeta_3$.

Since $x^t$ is a minimizer of $F(x)$ over the support set $S^t$, it immediately follows that

$$F(x^t) - F(x^{t-1}) \leq F(z_{S^t}^t) - F(x^{t-1}) \leq -\eta(1 - \eta\rho_{2s}^+) \left\| \nabla_{S^t\backslash S^{t-1}} F(x^{t-1}) \right\|^2.$$

Now we invoke Lemma 12 and pick $\eta \leq 1/\rho_{2s}^+$,

$$F(x^t) - F(x^{t-1}) \leq -2m\eta(1 - \eta\rho_{2s}^+) \cdot \frac{\left| S^t \backslash S^{t-1} \right|}{|S^t \backslash S^{t-1}| + |S \backslash S^{t-1}|} \left( F(x^{t-1}) - F(\bar{x}) \right),$$

which gives

$$F(x^t) - F(\bar{x}) \leq \mu_t \left( F(x^{t-1}) - F(\bar{x}) \right),$$

where $\mu_t = 1 - 2\eta\rho_{2s}^-(1 - \eta\rho_{2s}^+) \cdot \frac{|S^t\backslash S^{t-1}|}{|S^t\backslash S^{t-1}|+|S\backslash S^{t-1}|}$. $\quad\square$

**Lemma 12.** *Consider the PHT($r$) algorithm and assume $F(x)$ is $\rho_{k+s}^-$-RSC. Then for all $t \geq 1$,*

$$\left\| \nabla_{S^t\backslash S^{t-1}} F(x^{t-1}) \right\|^2 \geq 2\rho_{k+s}^- \delta_t \left( F(x^{t-1}) - F(\bar{x}) \right),$$

*where*

$$\delta_t = \frac{\left| S^t \backslash S^{t-1} \right|}{|S^t \backslash S^{t-1}| + |S \backslash S^{t-1}|}.$$

*Proof.* The lemma holds clearly for either $S^t = S^{t-1}$ or $F(\boldsymbol{x}^t) \leq F(\bar{\boldsymbol{x}})$. Hence, in the following we only prove the result by assuming $S^t \neq S^{t-1}$ and $F(\boldsymbol{x}^t) > F(\bar{\boldsymbol{x}})$. Due to the RSC property, we have

$$F(\bar{\boldsymbol{x}}) - F(\boldsymbol{x}^{t-1}) - \langle \nabla F(\boldsymbol{x}^{t-1}), \bar{\boldsymbol{x}} - \boldsymbol{x}^{t-1} \rangle \geq \frac{\bar{\rho}_{k+s}}{2} \left\| \bar{\boldsymbol{x}} - \boldsymbol{x}^{t-1} \right\|^2 ,$$

which implies

$$\langle \nabla F(\boldsymbol{x}^{t-1}), -\bar{\boldsymbol{x}} \rangle \geq \frac{\bar{\rho}_{k+s}}{2} \left\| \bar{\boldsymbol{x}} - \boldsymbol{x}^{t-1} \right\|^2 + F(\boldsymbol{x}^{t-1}) - F(\bar{\boldsymbol{x}})$$

$$\geq \sqrt{2\bar{\rho}_{k+s}} \left\| \bar{\boldsymbol{x}} - \boldsymbol{x}^{t-1} \right\| \sqrt{F(\boldsymbol{x}^{t-1}) - F(\bar{\boldsymbol{x}})}.$$

By invoking Lemma 13 with $\boldsymbol{u} = \nabla F(\boldsymbol{x}^{t-1})$ and $\boldsymbol{z} = -\bar{\boldsymbol{x}}$ therein, we have

$$\langle \nabla F(\boldsymbol{x}^{t-1}), -\bar{\boldsymbol{x}} \rangle \leq \sqrt{\frac{|S \setminus S^{t-1}|}{|S^t \setminus S^{t-1}|} + 1} \left\| \nabla_{S^t \setminus S^{t-1}} F(\boldsymbol{x}^{t-1}) \right\| \cdot \left\| \bar{\boldsymbol{x}}_{S \setminus S^{t-1}} \right\|$$

$$= \sqrt{\frac{|S \setminus S^{t-1}|}{|S^t \setminus S^{t-1}|} + 1} \left\| \nabla_{S^t \setminus S^{t-1}} F(\boldsymbol{x}^{t-1}) \right\| \cdot \left\| (\bar{\boldsymbol{x}} - \boldsymbol{x}^t)_{S \setminus S^{t-1}} \right\|$$

$$\leq \sqrt{\frac{|S \setminus S^{t-1}|}{|S^t \setminus S^{t-1}|} + 1} \left\| \nabla_{S^t \setminus S^{t-1}} F(\boldsymbol{x}^{t-1}) \right\| \cdot \left\| \bar{\boldsymbol{x}} - \boldsymbol{x}^t \right\| .$$

It is worth mentioning that the first inequality above holds because $\nabla F(\boldsymbol{x}^{t-1})$ is supported on $\overline{S^{t-1}}$ and $S^t \setminus S^{t-1}$ contains the $\left| S^t \setminus S^{t-1} \right|$ number of largest (in magnitude) elements of $\nabla F(\boldsymbol{x}^{t-1})$. Therefore, we obtain the result. $\qquad\square$

**Lemma 13** (Lemma 1 in [28]). *Let $\boldsymbol{u}$ and $\boldsymbol{z}$ be two distinct vectors and let $W = \mathrm{supp}\,(\boldsymbol{u}) \cap \mathrm{supp}\,(\boldsymbol{z})$. Also, let $U$ be the support set of the top $r$ (in magnitude) elements in $\boldsymbol{u}$. Then, the following holds for all $r \geq 1$:*

$$\langle \boldsymbol{u}, \boldsymbol{z} \rangle \leq \sqrt{\left\lceil \frac{|W|}{r} \right\rceil} \left\| \boldsymbol{u}_U \right\| \cdot \left\| \boldsymbol{z}_W \right\| .$$

## A.2 Proof of Theorem 3

*Proof.* Let $\rho^- := \rho_{2s+r}^-$ and $\rho^+ := \rho_{2s+r}^+$. Let $\phi := \phi_{2s+r} = 1 - \eta\rho^-$ be the quantity given in Lemma 17. Using Lemma 14, we obtain

$$\left\| \boldsymbol{x}^t - \bar{\boldsymbol{x}} \right\| \leq \left( \sqrt{2}\phi\kappa + \nu(\kappa - 1) \right) \left\| \boldsymbol{x}^{t-1} - \bar{\boldsymbol{x}} \right\| + \frac{2\nu + 4}{\rho^-} \left\| \nabla_{s+r} F(\bar{\boldsymbol{x}}) \right\| ,$$

where $\nu = \sqrt{s - r + 2}$. We need to ensure that the convergence coefficient is smaller than 1. Consider $\eta = \eta'/\rho^+$ with $\eta' \in (0, 1]$ for which $\phi = 1 - \eta'/\kappa$. It follows that

$$\sqrt{2}\phi\kappa + \nu(\kappa - 1) = \sqrt{2}(\kappa - \eta') + \nu(\kappa - 1) \leq (\sqrt{2} + \nu)(\kappa - \eta').$$

Hence, when we pick $1 - \frac{1}{\sqrt{2}+\nu} < \eta' \leq 1$, and the condition number satisfies

$$\kappa < \eta' + \frac{1}{\sqrt{2} + \nu},$$

the sequence of $\boldsymbol{x}^t - \bar{\boldsymbol{x}}$ contracts. On the other hand, using Lemma 19 we get

$$\left\| \boldsymbol{x}^t - \bar{\boldsymbol{x}} \right\| \leq \kappa \left\| \bar{\boldsymbol{x}}_{\overline{S^t}} \right\| + \frac{1}{\rho^-} \left\| \nabla_s F(\bar{\boldsymbol{x}}) \right\| .$$

Hence, applying Lemma 5 and Theorem 6 we obtain the result. $\qquad\square$

**Lemma 14.** *Consider the PHT($r$) algorithm with $k = s$. Suppose that $F(\boldsymbol{x})$ is $\rho_{2s+r}^-$-RSC and $\rho_{2s+r}^+$-RSS. Further suppose that $\kappa < 2$. Let the step size $\eta \leq 1/\rho_{2s+r}^+$. Then it holds that*

$$\left\|\boldsymbol{x}^t - \bar{\boldsymbol{x}}\right\| \leq \left(\sqrt{2}\phi\kappa + \nu(\kappa - 1)\right)\left\|\boldsymbol{x}^{t-1} - \bar{\boldsymbol{x}}\right\| + \frac{2\nu + 4}{\rho_{2s+r}^-}\left\|\nabla_{s+r}F(\bar{\boldsymbol{x}})\right\|,$$

*where $\phi = 1 - \eta\rho_{2s+r}^-$ and $\nu = \sqrt{s - r + 2}$.*

*Proof.* Consider the vector $\boldsymbol{z}_{J^t}^t$. It is easy to see that $J^t \setminus S^t$ contains the $r$ smallest elements of $\boldsymbol{z}_{J^t}^t$. Hence, for any subset $T \subset J^t$ such that $|T| \geq r$, we have

$$\left\|\boldsymbol{z}_{J^t \setminus S^t}^t\right\| \leq \left\|\boldsymbol{z}_T^t\right\|.$$

In particular, we choose $T = J^t \setminus S$ and obtain

$$\left\|\boldsymbol{z}_{J^t \setminus S^t}^t\right\| \leq \left\|\boldsymbol{z}_{J^t \setminus S}^t\right\|.$$

Eliminating the common contribution from $J^t \setminus (S^t \cup S)$ gives

$$\left\|\boldsymbol{z}_{J^t \cap S \setminus S^t}^t\right\| \leq \left\|\boldsymbol{z}_{J^t \cap S^t \setminus S}^t\right\|. \tag{13}$$

The LHS of (13) reads as

$$\begin{aligned}
\left\|\boldsymbol{z}_{J^t \cap S \setminus S^t}^t\right\| &= \left\|(\boldsymbol{x}^{t-1} - \eta\nabla F(\boldsymbol{x}^{t-1}) - \bar{\boldsymbol{x}})_{J^t \cap S \setminus S^t} + \bar{\boldsymbol{x}}_{J^t \setminus S^t}\right\| \\
&\geq \left\|\bar{\boldsymbol{x}}_{J^t \setminus S^t}\right\| - \left\|(\boldsymbol{x}^{t-1} - \eta\nabla F(\boldsymbol{x}^{t-1}) - \bar{\boldsymbol{x}})_{J^t \cap S \setminus S^t}\right\|,
\end{aligned}$$

while the RHS (13) is given by

$$\left\|\boldsymbol{z}_{J^t \cap S^t \setminus S}^t\right\| = \left\|(\boldsymbol{x}^{t-1} - \eta\nabla F(\boldsymbol{x}^{t-1}) - \bar{\boldsymbol{x}})_{J^t \cap S^t \setminus S}\right\|.$$

Hence, we have

$$\begin{aligned}
\left\|\bar{\boldsymbol{x}}_{J^t \setminus S^t}\right\| &\leq \left\|(\boldsymbol{x}^{t-1} - \eta\nabla F(\boldsymbol{x}^{t-1}) - \bar{\boldsymbol{x}})_{J^t \cap S \setminus S^t}\right\| + \left\|(\boldsymbol{x}^{t-1} - \eta\nabla F(\boldsymbol{x}^{t-1}) - \bar{\boldsymbol{x}})_{J^t \cap S^t \setminus S}\right\| \\
&\leq \sqrt{2}\left\|(\boldsymbol{x}^{t-1} - \eta\nabla F(\boldsymbol{x}^{t-1}) - \bar{\boldsymbol{x}})_{J^t}\right\| \\
&\leq \sqrt{2}\phi_{2s+r}\left\|\boldsymbol{x}^{t-1} - \bar{\boldsymbol{x}}\right\| + \sqrt{2}\eta\left\|\nabla_{k+r}F(\bar{\boldsymbol{x}})\right\|,
\end{aligned}$$

where we use Lemma 18 for the last inequality and $\phi_{2s+r} = 1 - \eta\rho_{2s+r}^-$ for $\eta \leq 1/\rho_{2s+r}^+$. On the other hand, Lemma 7 shows that

$$\left\|\bar{\boldsymbol{x}}_{\overline{J^t}}\right\| \leq \nu\left(1 - \frac{1}{\kappa}\right)\left\|\boldsymbol{x}^{t-1} - \bar{\boldsymbol{x}}\right\| + \frac{\nu}{\rho_{2s+r}^-}\left\|\nabla_{s+r}F(\bar{\boldsymbol{x}})\right\|,$$

where $\nu = \sqrt{s - r + 2}$. The fact $\overline{S^t} = (J^t \setminus S^t) \cup \overline{J^t}$ implies

$$\begin{aligned}
\left\|\bar{\boldsymbol{x}}_{\overline{S^t}}\right\| &\leq \left\|\bar{\boldsymbol{x}}_{J^t \setminus S^t}\right\| + \left\|\bar{\boldsymbol{x}}_{\overline{J^t}}\right\| \\
&\leq \left(\sqrt{2}\phi_{2s+r} + \nu\left(1 - \frac{1}{\kappa}\right)\right)\left\|\boldsymbol{x}^{t-1} - \bar{\boldsymbol{x}}\right\| + \left(\sqrt{2}\eta + \frac{\nu}{\rho_{2s+r}^-}\right)\left\|\nabla_{k+r}F(\bar{\boldsymbol{x}})\right\|.
\end{aligned}$$

Next, we invoke Lemma 19 to get

$$\left\|\boldsymbol{x}^t - \bar{\boldsymbol{x}}\right\| \leq \kappa\left\|\bar{\boldsymbol{x}}_{\overline{S^t}}\right\| + \frac{1}{\rho_{2s+r}^-}\left\|\nabla_k F(\bar{\boldsymbol{x}})\right\|.$$

Therefore,

$$\begin{aligned}
\left\|\boldsymbol{x}^t - \bar{\boldsymbol{x}}\right\| &\leq \left(\sqrt{2}\phi_{2s+r}\kappa + \nu(\kappa - 1)\right)\left\|\boldsymbol{x}^{t-1} - \bar{\boldsymbol{x}}\right\| + \left(\sqrt{2}\eta\kappa + \frac{\nu\kappa}{\rho_{2s+r}^-} + \frac{1}{\rho_{2s+r}^-}\right)\left\|\nabla_{s+r}F(\bar{\boldsymbol{x}})\right\| \\
&\leq \left(\sqrt{2}\phi_{2s+r}\kappa + \nu(\kappa - 1)\right)\left\|\boldsymbol{x}^{t-1} - \bar{\boldsymbol{x}}\right\| + \frac{2\nu + 4}{\rho_{2s+r}^-}\left\|\nabla_{s+r}F(\bar{\boldsymbol{x}})\right\|,
\end{aligned}$$

where we use the assumption that $\kappa < 2$ and $\eta \leq 1/\rho_{2s+r}^+ < 1/\rho_{2s+r}^-$ for the last inequality. $\qquad\square$

## A.3 Proof of Theorem 4

*Proof.* Using Lemma 15, we have

$$F(\boldsymbol{x}^t) - F(\bar{\boldsymbol{x}}) \le \mu \left( F(\boldsymbol{x}^{t-1}) - F(\bar{\boldsymbol{x}}) \right),$$

where

$$\mu = 1 - \frac{\eta \rho_{2k}^-(1 - \eta \rho_{2k}^+)}{2}.$$

Now Prop. 21 suggests that

$$\left\| \boldsymbol{x}^t - \bar{\boldsymbol{x}} \right\| \le \sqrt{2\kappa} \left( \sqrt{\mu} \right)^t \left\| \boldsymbol{x}^0 - \bar{\boldsymbol{x}} \right\| + \frac{3}{\rho_{2k}^-} \left\| \nabla_{k+s} F(\bar{\boldsymbol{x}}) \right\|,$$

and Lemma 19 implies

$$\left\| \boldsymbol{x}^t - \bar{\boldsymbol{x}} \right\| \le \kappa \left\| \bar{\boldsymbol{x}}_{\overline{S^t}} \right\| + \frac{1}{\rho_{2k}^-} \left\| \nabla_k F(\bar{\boldsymbol{x}}) \right\|.$$

Combining these with Lemma 5 and Theorem 6 we complete the proof. □

**Lemma 15.** *Consider the PHT(r) algorithm. Suppose that $F(\boldsymbol{x})$ is $\rho_{2k}^-$-RSC and $\rho_{2k}^+$-RSS, and let $\kappa = \rho_{2k}^+/\rho_{2k}^-$ be the condition number. Picking the step size $0 < \eta < 1/\rho_{2k}^+$ and the sparsity parameter $k \ge s + \left( 1 + \frac{4}{\eta^2 (\rho_{2k}^-)^2} \right) \min\{r, s\}$, then we have*

$$F(\boldsymbol{x}^t) - F(\boldsymbol{x}^{t-1}) \le -\frac{\eta \rho_{2k}^-(1 - \eta \rho_{2k}^+)}{2} \left( F(\boldsymbol{x}^{t-1}) - F(\bar{\boldsymbol{x}}) \right).$$

*Proof.* Using Lemma 16 we obtain

$$F(\boldsymbol{x}^t) - F(\boldsymbol{x}^{t-1}) \le -\frac{1 - \eta \rho_{2k}^+}{2\eta} \left\| \boldsymbol{z}_{S^t}^t - \boldsymbol{x}^{t-1} \right\|^2.$$

Note that for the right-hand side, we may expand it as follows:

$$\begin{aligned}
\left\| \boldsymbol{z}_{S^t}^t - \boldsymbol{x}^{t-1} \right\|^2 &= \left\| \boldsymbol{x}_{S^t}^{t-1} - \boldsymbol{x}^{t-1} - \eta \nabla_{S^t} F(\boldsymbol{x}^{t-1}) \right\|^2 \\
&= \left\| -\boldsymbol{x}_{S^{t-1} \setminus S^t}^{t-1} - \eta \nabla_{S^t \setminus S^{t-1}} F(\boldsymbol{x}^{t-1}) \right\|^2 \\
&= \left\| \boldsymbol{x}_{S^{t-1} \setminus S^t}^{t-1} \right\|^2 + \eta^2 \left\| \nabla_{S^t \setminus S^{t-1}} F(\boldsymbol{x}^{t-1}) \right\|^2,
\end{aligned}$$

where we use the fact that $\boldsymbol{x}^{t-1}$ is supported on $S^{t-1}$ and $\nabla F(\boldsymbol{x}^{t-1})$ is support on $\overline{S^{t-1}}$ for the second equality, and the third one follows in that the support sets are disjoint. It then follows quickly that

$$F(\boldsymbol{x}^t) - F(\boldsymbol{x}^{t-1}) \le -\frac{(1 - \eta \rho_{2k}^+)\eta}{2} \left\| \nabla_{S^t \setminus S^{t-1}} F(\boldsymbol{x}^{t-1}) \right\|^2.$$

It remains to lower bound the right-hand side in terms of $F(\boldsymbol{x}^{t-1}) - F(\bar{\boldsymbol{x}})$. In fact, in the following, we show that

$$\left\| \nabla_{S^t \setminus S^{t-1}} F(\boldsymbol{x}^{t-1}) \right\|^2 \ge \rho_{2k}^- \left( F(\boldsymbol{x}^{t-1}) - F(\bar{\boldsymbol{x}}) \right). \tag{14}$$

This suggests

$$F(\boldsymbol{x}^t) - F(\boldsymbol{x}^{t-1}) \le -\frac{\eta \rho_{2k}^-(1 - \eta \rho_{2k}^+)}{2} \left( F(\boldsymbol{x}^{t-1}) - F(\bar{\boldsymbol{x}}) \right)$$

which completes the proof. In the sequel, we prove the inequality (14) by discussing the size of the support set $S^t \setminus S^{t-1}$.

First, we consider $r \ge s$. Then it is possible that $\left| S^t \setminus S^{t-1} \right| \ge s$.

**Case 1.** $\left|S^t \setminus S^{t-1}\right| \geq s$. Using the RSC property, we have

$$
\begin{aligned}
\frac{\rho_{2k}^-}{2} &\left\|\bar{x} - x^{t-1}\right\|^2 \\
&\leq F(\bar{x}) - F(x^{t-1}) - \left\langle \nabla F(x^{t-1}), \bar{x} - x^{t-1}\right\rangle \\
&\leq F(\bar{x}) - F(x^{t-1}) + \frac{\rho_{2k}^-}{2}\left\|\bar{x} - x^{t-1}\right\|^2 + \frac{1}{2\rho_{2k}^-}\left\|\nabla_{S \cup S^{t-1}} F(x^{t-1})\right\|^2 \\
&= F(\bar{x}) - F(x^{t-1}) + \frac{\rho_{2k}^-}{2}\left\|\bar{x} - x^{t-1}\right\|^2 + \frac{1}{2\rho_{2k}^-}\left\|\nabla_{S \setminus S^{t-1}} F(x^{t-1})\right\|^2 .
\end{aligned}
$$

Therefore, we get

$$
\left\|\nabla_{S \setminus S^{t-1}} F(x^{t-1})\right\|^2 \geq 2\rho_{2k}^- \left(F(x^{t-1}) - F(\bar{x})\right).
$$

Recall that $S^t \setminus S^{t-1}$ contains the largest elements of $z^t_{\overline{S^{t-1}}}$. Hence, for any support set $T \subset \overline{S^{t-1}}$ with $|T| \leq \left|S^t \setminus S^{t-1}\right|$, we have

$$
\left\|z_T^t\right\| \leq \left\|z_{S^t \setminus S^{t-1}}^t\right\|.
$$

In particular, we can choose $T = S \setminus S^{t-1}$ as we assumed that $\left|S^t \setminus S^{t-1}\right| \geq s \geq |T|$. Then it holds that

$$
\left\|z_{S^t \setminus S^{t-1}}^t\right\|^2 \geq \left\|z_{S \setminus S^{t-1}}^t\right\|^2 .
$$

Note that for the left-hand side, $z_{S^t \setminus S^{t-1}}^t = -\eta \nabla_{S^t \setminus S^{t-1}} F(x^{t-1})$ while for the right-hand side, it is exactly equal to $-\eta \nabla_{S \setminus S^{t-1}} F(x^{t-1})$. This completes the proof of the first case.

**Case 2.** $\left|S^t \setminus S^{t-1}\right| < s \leq r$. The proof of this part is more involved. We still begin with the RSC property, which gives

$$
\begin{aligned}
\frac{\rho_{2k}^-}{2}\left\|\bar{x} - x^{t-1}\right\|^2 &\leq F(\bar{x}) - F(x^{t-1}) - \left\langle \nabla F(x^{t-1}), \bar{x} - x^{t-1}\right\rangle \\
&\leq F(\bar{x}) - F(x^{t-1}) + \frac{\rho_{2k}^-}{4}\left\|\bar{x} - x^{t-1}\right\|^2 + \frac{1}{\rho_{2k}^-}\left\|\nabla_{S \cup S^{t-1}} F(x^{t-1})\right\|^2 \\
&= F(\bar{x}) - F(x^{t-1}) + \frac{\rho_{2k}^-}{4}\left\|\bar{x} - x^{t-1}\right\|^2 + \frac{1}{\rho_{2k}^-}\left\|\nabla_{S \setminus S^{t-1}} F(x^{t-1})\right\|^2 \\
&= F(\bar{x}) - F(x^{t-1}) + \frac{\rho_{2k}^-}{4}\left\|\bar{x} - x^{t-1}\right\|^2 + \frac{1}{\rho_{2k}^-}\left\|\nabla_{S \setminus (S^t \cup S^{t-1})} F(x^{t-1})\right\|^2 \\
&\quad + \frac{1}{\rho_{2k}^-}\left\|\nabla_{(S^t \setminus S^{t-1}) \cap S} F(x^{t-1})\right\|^2 \\
&\leq F(\bar{x}) - F(x^{t-1}) + \frac{\rho_{2k}^-}{4}\left\|\bar{x} - x^{t-1}\right\|^2 + \frac{1}{\rho_{2k}^-}\left\|\nabla_{S \setminus (S^t \cup S^{t-1})} F(x^{t-1})\right\|^2 \\
&\quad + \frac{1}{\rho_{2k}^-}\left\|\nabla_{S^t \setminus S^{t-1}} F(x^{t-1})\right\|^2 . \quad\quad (15)
\end{aligned}
$$

Note that the last term is retained for deduction. What we need to show is a proper bound of the term $\left\|\nabla_{S \setminus (S^t \cup S^{t-1})} F(x^{t-1})\right\|^2$ above. First, we observe that

$$
z_{S \setminus (S^t \cup S^{t-1})}^t = -\eta \nabla_{S \setminus (S^t \cup S^{t-1})} F(x^{t-1}).
$$

Next, we compare the elements of $S \setminus (S^t \cup S^{t-1})$ to those in $(S^t \cap S^{t-1}) \setminus S$. For convenience, we denote $T = J^t \setminus (S^{t-1} \cup S^t)$. Since $S^t$ contains the $k$ largest elements of $z_{J^t}^t$, those of $(S^t \cap S^{t-1}) \setminus S$ are larger than those in $T$. On the other hand, recall that elements in $J^t \setminus S^{t-1}$ are larger than those in $\overline{J^t}$ due to the partial hard thresholding. Since $T$ is a subset of $J^t \setminus S^{t-1}$, we have that $T$ is larger

than $\overline{J^t}$. Consequently, elements in $(S^t \cap S^{t-1}) \setminus S$ are larger than those in $T \cup \overline{J^t} = \overline{S^{t-1} \cup S^t}$. This suggests that

$$\frac{\left\| z^t_{S \setminus (S^t \cup S^{t-1})} \right\|^2}{|S \setminus (S^t \cup S^{t-1})|} \leq \frac{\left\| z^t_{(S^t \cap S^{t-1}) \setminus S} \right\|^2}{|(S^t \cap S^{t-1}) \setminus S|}.$$

Note that $\left| S^t \setminus S^{t-1} \right| < s$ implies $\left| (S^t \cap S^{t-1}) \setminus S \right| \geq k - 2s$. Therefore,

$$\eta^2 \left\| \nabla_{S \setminus (S^t \cup S^{t-1})} F(x^{t-1}) \right\|^2 \leq \frac{s}{k-2s} \left\| x^{t-1}_{(S^t \cap S^{t-1}) \setminus S} - \eta \nabla_{(S^t \cap S^{t-1}) \setminus S} F(x^{t-1}) \right\|^2$$

$$= \frac{s}{k-2s} \left\| x^{t-1}_{(S^t \cap S^{t-1}) \setminus S} \right\|^2$$

$$= \frac{s}{k-2s} \left\| (x^{t-1} - \bar{x})_{(S^t \cap S^{t-1}) \setminus S} \right\|^2$$

$$\leq \frac{s}{k-2s} \left\| x^{t-1} - \bar{x} \right\|^2.$$

Plugging the above into (15), we obtain

$$\frac{\rho^-_{2k}}{2} \left\| \bar{x} - x^{t-1} \right\|^2 \leq F(\bar{x}) - F(x^{t-1}) + \frac{\rho^-_{2k}}{4} \left\| \bar{x} - x^{t-1} \right\|^2 + \frac{s}{(k-2s)\eta^2 \rho^-_{2k}} \left\| \bar{x} - x^{t-1} \right\|^2$$

$$+ \frac{1}{\rho^-_{2k}} \left\| \nabla_{S^t \setminus S^{t-1}} F(x^{t-1}) \right\|^2.$$

Picking $k \geq 2s + \frac{4s}{\eta^2 (\rho^-_{2k})^2}$ gives

$$\frac{\rho^-_{2k}}{2} \left\| \bar{x} - x^{t-1} \right\|^2 \leq F(\bar{x}) - F(x^{t-1}) + \frac{\rho^-_{2k}}{2} \left\| \bar{x} - x^{t-1} \right\|^2 + \frac{1}{\rho^-_{2k}} \left\| \nabla_{S^t \setminus S^{t-1}} F(x^{t-1}) \right\|^2,$$

which is exactly the claim (14).

Now we consider the parameter setting $r < s$. In this case, $\left| S^t \setminus S^{t-1} \right|$ cannot be greater than $s$. In fact, like we have done for Case 2, we can show that

$$\eta^2 \left\| \nabla_{S \setminus (S^t \cup S^{t-1})} F(x^{t-1}) \right\|^2 \leq \frac{r}{k-r-s} \left\| x^{t-1} - \bar{x} \right\|^2.$$

Plugging the above into (15), we obtain

$$\frac{\rho^-_{2k}}{2} \left\| \bar{x} - x^{t-1} \right\|^2 \leq F(\bar{x}) - F(x^{t-1}) + \frac{\rho^-_{2k}}{4} \left\| \bar{x} - x^{t-1} \right\|^2 + \frac{r}{(k-r-s)\eta^2 \rho^-_{2k}} \left\| \bar{x} - x^{t-1} \right\|^2$$

$$+ \frac{1}{\rho^-_{2k}} \left\| \nabla_{S^t \setminus S^{t-1}} F(x^{t-1}) \right\|^2.$$

Using $k \geq s + r + \frac{4r}{\eta^2 (\rho^-_{2k})^2}$ we prove (14).

Overall, we find that picking $k \geq s + \left( 1 + \frac{4}{\eta^2 (\rho^-_{2k})^2} \right) \min\{r, s\}$ always guarantees the result. $\qquad \square$

**Lemma 16.** *Consider the PHT$(r)$ algorithm. Suppose that $F(x)$ is $\rho^+_{2k}$-RSS. We have*

$$F(x^t) - F(x^{t-1}) \leq -\frac{1 - \eta \rho^+_{2k}}{2\eta} \left\| z^t_{S^t} - x^{t-1} \right\|^2.$$

*Proof.* We partition $z^t$ into four disjoint parts: $S^{t-1} \setminus S^t$, $S^{t-1} \cap S^t$, $S^t \setminus S^{t-1}$ and $\overline{J^t}$. It then follows that

$$\left\| z^t_{S^t} - z^t \right\|^2 = \left\| z^t_{S^{t-1} \setminus S^t} \right\|^2 + \left\| z^t_{\overline{J^t}} \right\|^2$$

$$\leq \left\| z^t_{S^t \setminus S^{t-1}} \right\|^2 + \left\| z^t_{\overline{J^t}} \right\|^2$$

$$= \left\| z^t_{\overline{S^{t-1}}} \right\|^2$$

$$= \eta^2 \left\| \nabla F(x^{t-1}) \right\|^2.$$

On the other hand, the LHS reads as

$$\left\| \mathbf{z}_{S^t}^t - \mathbf{z}^t \right\|^2 = \left\| \mathbf{z}_{S^t}^t - \mathbf{x}^{t-1} + \eta \nabla F(\mathbf{x}^{t-1}) \right\|^2$$

$$= \left\| \mathbf{z}_{S^t}^t - \mathbf{x}^{t-1} \right\|^2 + \eta^2 \left\| \nabla F(\mathbf{x}^{t-1}) \right\|^2 + 2\eta \left\langle \nabla F(\mathbf{x}^{t-1}), \mathbf{z}_{S^t}^t - \mathbf{x}^{t-1} \right\rangle.$$

Hence,

$$\left\langle \nabla F(\mathbf{x}^{t-1}), \mathbf{z}_{S^t}^t - \mathbf{x}^{t-1} \right\rangle \leq -\frac{1}{2\eta} \left\| \mathbf{z}_{S^t}^t - \mathbf{x}^{t-1} \right\|^2.$$

Using the RSS property, we have

$$F(\mathbf{x}^t) - F(\mathbf{x}^{t-1}) \leq F(\mathbf{y}^t) - F(\mathbf{x}^{t-1})$$

$$= F(\mathbf{z}_{S^t}^t) - F(\mathbf{x}^{t-1})$$

$$\leq \left\langle \nabla F(\mathbf{x}^{t-1}), \mathbf{z}_{S^t}^t - \mathbf{x}^{t-1} \right\rangle + \frac{\rho_{2k}^+}{2} \left\| \mathbf{z}_{S^t}^t - \mathbf{x}^{t-1} \right\|^2$$

$$\leq -\frac{1 - \eta \rho_{2k}^+}{2\eta} \left\| \mathbf{z}_{S^t}^t - \mathbf{x}^{t-1} \right\|^2.$$

$\square$

## B  Technical Lemmas

**Lemma 17.** *Suppose that $F(\mathbf{x})$ is $\rho_K^-$-RSC and $\rho_K^+$-RSS for some sparsity level $K > 0$. Then for all $\theta \in \mathbb{R}$, all vectors $\mathbf{x}, \mathbf{x}' \in \mathbb{R}^d$ and for any Hessian matrix $\mathbf{H}$ of $F(\mathbf{x})$, we have*

$$|\langle \mathbf{x}, (\mathbf{I} - \theta \mathbf{H})\mathbf{x}' \rangle| \leq \phi_K \|\mathbf{x}\| \cdot \|\mathbf{x}'\|,$$

*provided that $|\operatorname{supp}(\mathbf{x}) \cup \operatorname{supp}(\mathbf{x}')| \leq K$, and*

$$\|((\mathbf{I} - \theta \mathbf{H})\mathbf{x})_S\| \leq \phi_K \|\mathbf{x}\|, \quad \text{if } |S \cup \operatorname{supp}(\mathbf{x})| \leq K,$$

*where*

$$\phi_K = \max\left\{ \left|\theta \rho_K^- - 1\right|, \left|\theta \rho_K^+ - 1\right| \right\}.$$

*Proof.* Since $\mathbf{H}$ is a Hessian matrix, we always have a decomposition $\mathbf{H} = \mathbf{A}^\top \mathbf{A}$ for some matrix $\mathbf{A}$. Denote $T = \operatorname{supp}(\mathbf{x}) \cup \operatorname{supp}(\mathbf{x}')$. By simple algebra, we have

$$|\langle \mathbf{x}, (\mathbf{I} - \theta \mathbf{H})\mathbf{x}' \rangle| = |\langle \mathbf{x}, \mathbf{x}' \rangle - \theta \langle \mathbf{A}\mathbf{x}, \mathbf{A}\mathbf{x}' \rangle|$$

$$\overset{\zeta_1}{=} |\langle \mathbf{x}, \mathbf{x}' \rangle - \theta \langle \mathbf{A}_T \mathbf{x}, \mathbf{A}_T \mathbf{x}' \rangle|$$

$$= \left| \left\langle \mathbf{x}, (\mathbf{I} - \theta \mathbf{A}_T^\top \mathbf{A}_T)\mathbf{x}' \right\rangle \right|$$

$$\leq \left\| \mathbf{I} - \theta \mathbf{A}_T^\top \mathbf{A}_T \right\| \cdot \|\mathbf{x}\| \cdot \|\mathbf{x}'\|$$

$$\overset{\zeta_2}{\leq} \max\left\{ \left|\theta \rho_K^- - 1\right|, \left|\theta \rho_K^+ - 1\right| \right\} \cdot \|\mathbf{x}\| \cdot \|\mathbf{x}'\|.$$

Here, $\zeta_1$ follows from the fact that $\operatorname{supp}(\mathbf{x}) \cup \operatorname{supp}(\mathbf{y}) = T$ and $\zeta_2$ holds because the RSC and RSS properties imply that the singular values of any Hessian matrix restricted on an $K$-sparse support set are lower and upper bounded by $\rho_K^-$ and $\rho_K^+$, respectively.

For some index set $S$ subject to $|S \cup \operatorname{supp}(\mathbf{x})| \leq K$, let $\mathbf{x}' = ((\mathbf{I} - \theta \mathbf{H})\mathbf{x})_S$. We immediately obtain

$$\|\mathbf{x}'\|^2 = \langle \mathbf{x}', (\mathbf{I} - \theta \mathbf{H})\mathbf{x} \rangle \leq \phi_K \|\mathbf{x}'\| \cdot \|\mathbf{x}\|,$$

indicating

$$\|((\mathbf{I} - \theta \mathbf{H})\mathbf{x})_S\| \leq \phi_K \|\mathbf{x}\|.$$

$\square$

**Lemma 18.** *Suppose that $F(\boldsymbol{x})$ is $\rho_K^-$-RSC and $\rho_K^+$-RSS for some sparsity level $K > 0$. For all vectors $\boldsymbol{x}, \boldsymbol{x}' \in \mathbb{R}^d$ and support set $T$ such that $|\operatorname{supp}(\boldsymbol{x} - \boldsymbol{x}') \cup T| \leq K$, the following holds for all $\theta \in \mathbb{R}$:*

$$\|(\boldsymbol{x} - \boldsymbol{x}' - \theta \nabla F(\boldsymbol{x}) + \theta \nabla F(\boldsymbol{x}'))_T\| \leq \phi_K \|\boldsymbol{x} - \boldsymbol{x}'\|,$$

*where $\phi_K$ is given in Lemma 17.*

*Proof.* In fact, for any two vectors $\boldsymbol{x}$ and $\boldsymbol{x}'$, there always exists a quantity $t \in [0, 1]$, such that

$$\nabla F(\boldsymbol{x}) - \nabla F(\boldsymbol{x}') = \nabla^2 F(t\boldsymbol{x} + (1 - t)\boldsymbol{x}')(\boldsymbol{x} - \boldsymbol{x}').$$

Let $\boldsymbol{H} = \nabla^2 F(t\boldsymbol{x} + (1 - t)\boldsymbol{x}')$. We write

$$
\begin{aligned}
&\|(\boldsymbol{x} - \boldsymbol{x}' - \theta \nabla F(\boldsymbol{x}) + \theta \nabla F(\boldsymbol{x}'))_T\| \\
&= \|(\boldsymbol{x} - \boldsymbol{x}' - \theta \boldsymbol{H}(\boldsymbol{x} - \boldsymbol{x}'))_T\| \\
&= \|((\boldsymbol{I} - \theta \boldsymbol{H})(\boldsymbol{x} - \boldsymbol{x}'))_T\| \\
&\leq \phi_K \|\boldsymbol{x} - \boldsymbol{x}'\|,
\end{aligned}
$$

where the last inequality applies Lemma 17. $\qquad\square$

**Lemma 19.** *Suppose that $F(\boldsymbol{x})$ is $\rho_K^-$-RSC and $\rho_K^+$-RSS for some sparsity level $K > 0$. Let $\kappa := \rho_K^+/\rho_K^-$. For all vectors $\boldsymbol{x}, \boldsymbol{x}' \in \mathbb{R}^d$ with $|\operatorname{supp}(\boldsymbol{x}) \cup \operatorname{supp}(\boldsymbol{x}')| \leq K$, we have*

$$\|\boldsymbol{x} - \boldsymbol{x}'\| \leq \kappa \|\boldsymbol{x}'_{\overline{T}}\| + \frac{1}{\rho_K^-} \|(\nabla F(\boldsymbol{x}) - \nabla F(\boldsymbol{x}'))_T\|,$$

$$\|(\boldsymbol{x} - \boldsymbol{x}')_T\| \leq \left(1 - \frac{1}{\kappa}\right) \|\boldsymbol{x} - \boldsymbol{x}'\| + \frac{1}{\rho_K^-} \|(\nabla F(\boldsymbol{x}) - \nabla F(\boldsymbol{x}'))_T\|.$$

*where $T$ is the support set of $\boldsymbol{x}$.*

*Proof.* We begin with bounding the $\ell_2$-norm of the difference of $\boldsymbol{x}$ and $\boldsymbol{x}'$. Let $\Omega = \operatorname{supp}(\boldsymbol{x}')$. For any positive scalar $\theta \in \mathbb{R}$ we have

$$
\begin{aligned}
\|(\boldsymbol{x} - \boldsymbol{x}')_T\|^2 &= \langle \boldsymbol{x} - \boldsymbol{x}' - \theta \nabla F(\boldsymbol{x}) + \theta \nabla F(\boldsymbol{x}'), (\boldsymbol{x} - \boldsymbol{x}')_T \rangle \\
&\quad + \theta \langle \nabla F(\boldsymbol{x}) - \nabla F(\boldsymbol{x}'), (\boldsymbol{x} - \boldsymbol{x}')_T \rangle \\
&\leq \|(\boldsymbol{x} - \boldsymbol{x}' - \theta \nabla F(\boldsymbol{x}) + \theta \nabla F(\boldsymbol{x}'))_T\| \cdot \|(\boldsymbol{x} - \boldsymbol{x}')_T\| \\
&\quad + \theta \|(\nabla F(\boldsymbol{x}) - \nabla F(\boldsymbol{x}'))_T\| \cdot \|(\boldsymbol{x} - \boldsymbol{x}')_T\| \\
&\leq \|\boldsymbol{x} - \boldsymbol{x}' - \theta (\nabla F(\boldsymbol{x}))_{T \cup \Omega} + \theta (\nabla F(\boldsymbol{x}'))_{T \cup \Omega}\| \cdot \|(\boldsymbol{x} - \boldsymbol{x}')_T\| \\
&\quad + \theta \|(\nabla F(\boldsymbol{x}) - \nabla F(\boldsymbol{x}'))_T\| \cdot \|(\boldsymbol{x} - \boldsymbol{x}')_T\| \\
&\leq \phi_K \|\boldsymbol{x} - \boldsymbol{x}'\| \cdot \|(\boldsymbol{x} - \boldsymbol{x}')_T\| + \theta \|(\nabla F(\boldsymbol{x}) - \nabla F(\boldsymbol{x}'))_T\| \cdot \|(\boldsymbol{x} - \boldsymbol{x}')_T\|,
\end{aligned}
$$

where we recall that $\phi_K$ is given in Lemma 17. Dividing both sides by $\|(\boldsymbol{x} - \bar{\boldsymbol{x}})_T\|$ gives

$$\|(\boldsymbol{x} - \boldsymbol{x}')_T\| \leq \phi_K \|\boldsymbol{x} - \boldsymbol{x}'\| + \theta \|(\nabla F(\boldsymbol{x}) - \nabla F(\boldsymbol{x}'))_T\|.$$

On the other hand,

$$
\begin{aligned}
\|\boldsymbol{x} - \boldsymbol{x}'\| &\leq \|(\boldsymbol{x} - \boldsymbol{x}')_T\| + \|(\boldsymbol{x} - \boldsymbol{x}')_{\overline{T}}\| \\
&\leq \phi_K \|\boldsymbol{x} - \boldsymbol{x}'\| + \theta \|(\nabla F(\boldsymbol{x}) - \nabla F(\boldsymbol{x}'))_T\| + \|\boldsymbol{x}'_{\overline{T}}\|.
\end{aligned}
$$

Hence, we have

$$\|\boldsymbol{x} - \boldsymbol{x}'\| \leq \frac{1}{1 - \phi_K} \|\boldsymbol{x}'_{\overline{T}}\| + \frac{\theta}{1 - \phi_K} \|(\nabla F(\boldsymbol{x}) - \nabla F(\boldsymbol{x}'))_T\|.$$

Picking $\theta = 1/\rho_K^+$, we have $\phi_K = 1 - \frac{1}{\kappa}$. Plugging these into the above and noting that $\rho_K^+ \geq \rho_K^-$ complete the proof. $\qquad\square$

**Lemma 20.** *Suppose that $F(\boldsymbol{x})$ is $\rho_K^-$-RSC. Then for any vectors $\boldsymbol{x}$ and $\boldsymbol{x}'$ with $\|\boldsymbol{x} - \boldsymbol{x}'\|_0 \leq K$, the following holds:*

$$\|\boldsymbol{x} - \boldsymbol{x}'\| \leq \sqrt{\frac{2\max\{F(\boldsymbol{x}) - F(\boldsymbol{x}'), 0\}}{\rho_K^-}} + \frac{2\|(\nabla F(\boldsymbol{x}'))_T\|}{\rho_K^-},$$

*where $T = \mathrm{supp}\,(\boldsymbol{x} - \boldsymbol{x}')$.*

*Proof.* The RSC property immediately implies

$$F(\boldsymbol{x}) - F(\boldsymbol{x}') \geq \langle \nabla F(\boldsymbol{x}'), \boldsymbol{x} - \boldsymbol{x}' \rangle + \frac{\rho_K^-}{2}\|\boldsymbol{x} - \boldsymbol{x}'\|^2$$

$$\geq -\|\nabla_T F(\boldsymbol{x}')\| \cdot \|\boldsymbol{x} - \boldsymbol{x}'\| + \frac{\rho_K^-}{2}\|\boldsymbol{x} - \boldsymbol{x}'\|^2.$$

Discussing the sign of $F(\boldsymbol{x}) - F(\boldsymbol{x}')$ and solving the above quadratic inequality completes the proof. $\square$

**Proposition 21.** *Suppose that $F(\boldsymbol{x})$ is $\rho_{k+s}^-$-RSC and $\rho_{2k}^+$-RSS. Let $\kappa := \rho_{2k}^+/\rho_{k+s}^-$. Suppose that for all $t \geq 1$, $\boldsymbol{x}^t$ is $k$-sparse and the following holds:*

$$F(\boldsymbol{x}^t) - F(\bar{\boldsymbol{x}}) \leq \mu_t \left( F(\boldsymbol{x}^{t-1}) - F(\bar{\boldsymbol{x}}) \right) + \tau,$$

*where $0 < \mu_t < \mu < 1$ for some $\mu$, $\tau \geq 0$ and $\bar{\boldsymbol{x}}$ is an arbitrary $s$-sparse signal. Then,*

$$\left\|\boldsymbol{x}^t - \bar{\boldsymbol{x}}\right\| \leq \sqrt{2\kappa}(\sqrt{\mu_1\mu_2 \ldots \mu_t})\left\|\boldsymbol{x}^0 - \bar{\boldsymbol{x}}\right\| + \frac{3}{\rho_{k+s}^-}\|\nabla_{k+s}F(\bar{\boldsymbol{x}})\| + \sqrt{\frac{2\tau}{\rho_{k+s}^-(1-\mu)}}.$$

*Proof.* The RSS property implies that

$$F(\boldsymbol{x}^0) - F(\bar{\boldsymbol{x}}) \leq \langle \nabla F(\bar{\boldsymbol{x}}), \boldsymbol{x}^0 - \bar{\boldsymbol{x}} \rangle + \frac{\rho_{2k}^+}{2}\left\|\boldsymbol{x}^0 - \bar{\boldsymbol{x}}\right\|^2$$

$$\leq \frac{\rho_{2k}^+}{2}\left\|\boldsymbol{x}^0 - \bar{\boldsymbol{x}}\right\|^2 + \frac{1}{2\rho_{2k}^+}\|\nabla_{k+s}F(\bar{\boldsymbol{x}})\|^2 + \frac{\rho_{2k}^+}{2}\left\|\boldsymbol{x}^0 - \bar{\boldsymbol{x}}\right\|^2$$

$$\leq \rho_{2k}^+\left\|\boldsymbol{x}^0 - \bar{\boldsymbol{x}}\right\|^2 + \frac{1}{2\rho_{2k}^+}\|\nabla_{k+s}F(\bar{\boldsymbol{x}})\|^2.$$

Denote $\mu_{1:t} = \mu_1\mu_2 \ldots \mu_t$. We obtain

$$F(\boldsymbol{x}^t) - F(\bar{\boldsymbol{x}}) \leq \mu_{1:t}\rho^+\left\|\boldsymbol{x}^0 - \bar{\boldsymbol{x}}\right\|^2 + \frac{1}{2\rho_{2k}^+}\|\nabla_{k+s}F(\bar{\boldsymbol{x}})\|^2 + \frac{\tau}{1-\mu}.$$

By Lemma 20, we have

$$\left\|\boldsymbol{x}^t - \bar{\boldsymbol{x}}\right\|$$

$$\leq \sqrt{\frac{2}{\rho_{k+s}^-}}\sqrt{\mu_{1:t}\rho_{2k}^+\left\|\boldsymbol{x}^0 - \bar{\boldsymbol{x}}\right\|^2 + \frac{1}{2\rho_{2k}^+}\|\nabla_{k+s}F(\bar{\boldsymbol{x}})\|^2 + \frac{\tau}{1-\mu}} + \frac{2}{\rho_{k+s}^-}\|\nabla_{k+s}F(\bar{\boldsymbol{x}})\|$$

$$\leq \sqrt{2\kappa}(\sqrt{\mu_{1:t}})\left\|\boldsymbol{x}^0 - \bar{\boldsymbol{x}}\right\| + \sqrt{\frac{1}{\rho_{k+s}^-\rho_{2k}^+}}\|\nabla_{k+s}F(\bar{\boldsymbol{x}})\| + \frac{2}{\rho_{k+s}^-}\|\nabla_{k+s}F(\bar{\boldsymbol{x}})\| + \sqrt{\frac{2\tau}{\rho_{k+s}^-(1-\mu)}}$$

$$\leq \sqrt{2\kappa}(\sqrt{\mu_{1:t}})\left\|\boldsymbol{x}^0 - \bar{\boldsymbol{x}}\right\| + \frac{3}{\rho_{k+s}^-}\|\nabla_{k+s}F(\bar{\boldsymbol{x}})\| + \sqrt{\frac{2\tau}{\rho_{k+s}^-(1-\mu)}}.$$

$\square$