[Reviews · NeurIPS 2017]

Reviewer 1



Summary. This paper studies support recovery of the partial hard thresholding algorithm, and have derived, under certain conditions, the iteration complexity of the partial hard thresholding algorithm. Quality. There seems no explicit description about the condition under which the PHT(r) algorithm terminates, which makes the statement of Proposition 1 difficult to understand. It would have been better if concrete arguments on specific problem examples were given. They are presented only in a very brief and abstract manner in the last paragraph of Section 2. In particular, if nonzeros in \bar{x} are distributed according to a distribution without discontinuity at x=0, then x_min should scale as O(1/n), so that it should become difficult to satisfy the x_min condition when n is large. COMMENT AFTER REBUTTAL: In the above, I should have written that x_min should scale as O(1/d). I am pretty sorry for my careless mistake. Clarity. I think that the description of the simulations is so brief that one cannot relate the simulation setups with the theoretical results. More concretely, the function F(x) adopted in the simulations should be explicitly stated, as well as the condition numbers and other parameters appearing in the theoretical results. Of interest also would be whether the 10,000 iterations are sufficient with regard to the theoretical guarantees. Also, nonzeros of the signals are generated as Gaussians, so that x_min values vary from trial to trial, as well as the s values specified. Lemma 4 and Theorem 5 in the main text appear in the supplementary material as Lemma 19 and Theorem 20 without explicit statement that they are the same. Originality. I think that this work would be moderately original, in that it seems that it has extended the existing arguments on support recovery via hard thresholding to partial hard thresholding, which would certainly be non-trivial. Significance. Since the partial hard thresholding includes the conventional hard thresholding as well as what is called the orthogonal matching pursuit with replacement as special cases, the theoretical support-recovery guarantees for the partial hard thresholding algorithm presented here should be of significance. Minor points: Line 16: ha(ve -> s) found Line 56: in turn indicate(s) Line 76: (Lowercase -> A lowercase) letter(s) Line 132: restricted (strongly) smooth Line 196: The(n) the support Line 287: i.i.d. (standard) normal variables. Line 304: significantly reduce(s)

Reviewer 2



Review prior to rebuttal: The paper provides analytical results on partial hard thresholding for sparse signal recovery that are based on RIP and on an alternative condition (RSC and RSS) and that is applicable to arbitrary signals. The description of previous results is confusing at times. First, [4] has more generic results than indicated in this paper in line 51. Remark 7 provides a result for arbitrary signals, while Theorems 5 and 6 consider all sufficiently sparse signals. Second, [13] provides a guarantee for sparse signal recovery (Theorem 4), and so it is not clear what the authors mean by “parameter estimation only” on line 59. If exact recovery is achieved, it is obvious that the support is estimated correctly too. Third, it is also not clear why Theorem 2 is described as an RIP-based guarantee - how does RIP connect to the condition number, in particular since RIP is a property that can go beyond specific matrix constructions? The simulation restricts itself to s-sparse signals, and so it is not clear that the authors are testing the aspects of the results that go beyond those available in existing work. It is also not clear how the results test the dependence on the condition number of the matrix. Minor comments follow. Title: “A Towards” is not grammatically correct. Line 128: the names of the properties are grammatically incorrect: convex -> convexity, smooth -> smoothness? or add “property” at the end of each? Line 136: calling M/m “the condition number of the problem” can be confusing, since this terminology is already used in linear algebra. Is there a relationship between these two quantities? Line 140: there is no termination condition given in the algorithm. Does this mean when the gradient in the first step is equal to zero? When the support St does not change in consecutive iterations? Line 185: the paper refers to a comparison of the analysis of [4] being “confined to a special signal” versus the manuscript’s “generalization… [to] a family of algorithms”. It is not clear how these two things compare to one another. Line 301: degrade -> degradation Line 304: reduces -> reduce Response to rebuttal: The authors have addressed some of the points above (in particular relevant to the description of prior work), but the impact of the contribution appears to be "borderline" for acceptance. Given that the paper is acceptable if the authors implement the changes described in the rebuttal, I have set my score to "marginally above threshold".

Reviewer 3



This paper analyzes the support recovery performance of the PHT algrorithm of Jain et al. While in Jain et al.'s paper the main focus (at least with respect to the PHT family) was estimation error, the current paper aims to extend this to support recovery performance. The technical contribution of the paper appears sound to me. However, its not entirely clear how different the techniques are to those used in [4]. I am willing to revise my rating if the authors can make this more clear. The presentation of the paper needs improvement. - There are several typos throughout. A thorough pass needs to be made. - The title of the paper is grammatically incorrect -- perhaps the authors meant "Towards a Unified Analysis of Support Recovery". This phrase "a towards unified analysis" is used in other places in the manuscript, and its incorrect. - In the abstract the authors use \kappa and say that it is THE condition number, while it is unclear for a while which condition number it is. This should be replaced by something like "where \kappa is an approporiate condition number", since the authors probably dont want to introduce the formal definition in the abstract.